# Convergent evolution of hexenal isomerases in Lepidoptera and plants

Yu-Hsien Lin [1,2] ✉, Bulah Chia-hsiang Wu [3], Abdoallah Sharaf [4], Sophie Maartje Elisabeth Heijblom [1], Ilias Prattis [1], Ching-Wen Tan[5,6], Rudolf J. Schilder [6,7], Jared Gregory Ali [6] & Silke Allmann [1] ✉

Green leaf volatiles (GLVs) are six-carbon volatile organic compounds that mediate plant responses to environmental stresses. The quantity and composition of emitted GLVs can vary with stress type, allowing plants to fine-tune their volatile blends. In addition, insect herbivores are capable of modulating these emissions. A key mechanism underlying this plasticity is the conversion of $Z$-3-hexenal to $E$-2-hexenal by the enzyme (3$Z$):(2$E$)-hexenal isomerase (Hi), which reshapes GLV profiles and may influence multitrophic interactions. Here we investigate the evolutionary origin, functional diversification and catalytic mechanisms of lepidopteran Hi homologues, which belong to the glucose–methanol–choline oxidoreductase family. Phylogenetic analysis of 34 lepidopteran species identified a distinct glucose–methanol–choline-β subclade enriched in Hi homologues, largely confined to the Apoditrysia lineage. Functional assays showed species-specific variation in Hi activity, with *Manduca sexta* Hi-1 displaying the highest activity among tested homologues under identical protein concentrations, both in vitro and in planta. Structural modelling and site-directed mutagenesis revealed that Hi activity requires a flavin adenine dinucleotide cofactor enabling the identification of key residues critical for flavin adenine dinucleotide binding. Comparative phylogenetics further suggests that Hi enzymes in plants and Lepidoptera evolved independently from unrelated enzyme families, representing a case of functional convergence coinciding with the Cretaceous angiosperm radiation.

Plants interact with their environment through the emission of volatile organic compounds[1]. Among these, green leaf volatiles (GLVs) form a group of six-carbon (C₆) molecules that impart the 'grassy' scent to foliage[2]. Produced by most green plants, GLVs are emitted within seconds of mechanical wounding, herbivore feeding or various abiotic stresses[3]. They contribute to a plant's direct defence, by deterring herbivores and inhibiting pathogenic infection[4,5], and to a plant's indirect defence, by attracting natural enemies of herbivores[6–8]. Owing to this

near-instantaneous release, GLVs also serve as ecological 'alarm signals' that prime and, in some cases, directly trigger plant defences in neighbouring tissues and nearby plants[9–12].

The biosynthesis of GLVs is initiated from polyunsaturated fatty acids, primarily α-linolenic and linoleic acid, via the enzymatic action of lipoxygenases (LOX)[13]. LOXs oxygenate these fatty acids, forming hydroperoxides, subsequently cleaved by hydroperoxide lyases into $Z$-3-hexenal (Z3AL) or hexanal, depending on the initial fatty acid

[1]Green Life Sciences Research Cluster, Department of Plant Physiology, Swammerdam Institute for Life Sciences, University of Amsterdam, Amsterdam, the Netherlands. [2]Master Program in Global Agriculture Technology and Genomic Science, International College, National Taiwan University, Taipei, Taiwan. [3]Biology Centre of the Czech Academy of Sciences, Institute of Entomology, Ceske Budejovice, Czechia. [4]SequAna Core Facility, Department of Biology, University of Konstanz, Konstanz, Germany. [5]Department of Entomology, National Chung Hsing University, Taichung, Taiwan. [6]Department of Entomology, The Pennsylvania State University, University Park, PA, USA. [7]Department of Biology, The Pennsylvania State University, University Park, PA, USA. ✉e-mail: yuhsienlin@ntu.edu.tw; S.Allmann@uva.nl

substrate. Within the GLV biosynthesis pathway, the rearrangement of Z3AL to *E*-2-hexenal (E2AL) is particularly important because it reshapes the *Z*3/*E*2 ratio and thereby modulates the downstream formation of the corresponding alcohols and acetates[14]. Although Z3AL is emitted first, E2AL exhibits stronger antimicrobial activity and induces pronounced downstream responses, which may relate to its higher reactivity towards nucleophilic biomolecules[15–17]. For example, recent studies showed that E2AL triggers a calcium burst and activates the WRKY46–MYC2 transcriptional module in *Arabidopsis*, thereby promoting flavonoid accumulation and enhancing anti-herbivore defences[18]. This rearrangement from Z3AL to E2AL could occur spontaneously due to the intrinsic instability of Z3AL but becomes more efficient when catalysed by the (3*Z*):(2*E*)-hexenal isomerase (Hi). Plant-derived HI enzymes have been identified in various species, including cucumber, tomato and rice, and belong to the cupin superfamily[19–22]. For clarity, we refer to plant hexenal isomerase as 'HI', whereas lepidopteran hexenal isomerase and enzymatic activity are denoted as 'Hi' throughout this study.

Intriguingly, larvae of the hawk moth (*Manduca sexta*) produce a functionally analogous but phylogenetically distinct Hi protein in their oral secretions (OS), affiliated with the glucose–methanol–choline (GMC) oxidoreductase that converts plant-produced Z3AL into E2AL while feeding[23]. Such conversion alters the ratio of *Z*-3/*E*-2-GLVs, which guides female moths in choosing oviposition sites[24] but paradoxically also serves as the cue that attracts their natural enemies[25]. Although Z3AL to E2AL conversion is the well-characterized reaction, both lepidopteran and plant Hi can also act on related *Z*-3-alkenals such as *Z*-3-octenal or *Z*-3-nonenal and may fulfil physiological roles in Lepidoptera in addition to their ecological functions[20,23]. Hi activity independently emerged in plants and Lepidoptera via distinct protein families, representing a compelling case of convergent evolution. In vitro assays reveal that Hi activity varies among lepidopteran species[23]. Semi-field trials confirm this pattern: *M. sexta* releases a pronounced burst of E2AL when feeding on solanaceous hosts, whereas *Chloridea virescens* produces only a negligible Z3AL to E2AL conversion[26].

Despite this ecological and evolutionary importance, the origins and functional diversity of Hi are still poorly understood in Lepidoptera. Only a single Hi protein from *M. sexta* has been biochemically and functionally characterized to date[23], and interspecies variation in Hi activity remains unexplored. In this study, we combine phylogenetic and functional analysis to reconstruct the evolutionary history of lepidopteran Hi. We map its taxonomic distribution and compare the enzymatic activity of Hi homologues from multiple lepidopteran taxa. We also identify the flavin adenine dinucleotide (FAD)-binding and catalytic motifs that are crucial for lepidopteran Hi activity. Finally, we chart Hi evolution across 34 lepidopteran species and 183 species in the green lineage (Viridiplantae), revealing that both plant and lepidopteran Hi arose during the Cretaceous angiosperm radiation.

## Results

### Phylogenetic and functional analysis of putative (3*Z*):(2*E*)-Hi genes in Lepidoptera

The first lepidopteran Hi characterized in our earlier study[23], *M. sexta* Hi (MsHi-1), is a member of the GMC oxidoreductase family. This protein family is defined by conserved N-terminal (PF00732) and C-terminal (PF05199) domains. To identify related homologues in Lepidoptera, we used profile hidden Markov models (HMMs) based on these domains. Subsequently, we constructed a maximum-likelihood phylogram comprising 1,251 GMC oxidoreductases from 34 lepidopteran species, spanning both non-Ditrysia and Ditrysia lineages (Supplementary Table 1). Consistent with previous studies[27,28], the lepidopteran GMC genes grouped into distinct subfamilies, showing substantial expansion within the GMC-β subfamily (Fig. 1a). MsHi-1, along with its gene duplicate lacking Hi activity, MsHi-like, clustered

within a well-supported subclade (bootstrap value of 100) within the GMC-β subfamily (Fig. 1a and Supplementary Fig. 1, highlighted in orange). To determine whether this subclade might represent a broader set of Hi homologues, we selected all homologous genes from four species—*Bombyx mori*, *M. sexta*, *C. virescens* and *Danaus plexippus*—that cluster within a monophyletic group (bootstrap value of 84) containing MsHi-1 (XP_030035814) (Supplementary Fig. 2), for further functional characterization. OS of these four species had previously been shown to exhibit Hi activity[23].

Since the salivary glands and midgut are primary sources of enzymes in OS, we first investigated whether the selected genes potentially encode Hi enzymes in OS by comparing their expression levels in the salivary glands and midgut to the non-OS source tissue, fat body. The majority of candidate genes were highly expressed in the salivary glands, whereas MsHi-like in *M. sexta* and DpHi-1 in *D. plexippus* showed comparatively higher expression in the midgut (Fig. 1b–e).

To assess the Hi activity of these putative homologues, we cloned the corresponding cDNAs into pGEX vectors and expressed the recombinant proteins in *Escherichia coli* BL21 (Supplementary Fig. 3). A sole homologue from *Pieris rapae*, present in the putative Hi clade (Supplementary Fig. 2), was included as a negative control, as previous studies have shown no Hi activity in the OS of this species[23,29]. After expression and purification, we tested three protein quantities (0.01, 0.1 and 1 μg) for their ability to convert Z3AL into E2AL in vitro using solid-phase microextraction (SPME)–gas chromatography (GC)–quadrupole time-of-flight mass spectrometry (qToF-MS). At least one protein from each species showed a concentration-dependent increase in Hi activity (Fig. 2a). In *M. sexta*, a second homologue (MsHi-2) also displayed Hi activity, albeit at lower levels than MsHi-1. Representative chromatograms of the 1 μg protein reactions confirm E2AL formation in active homologues (Fig. 2b). As expected, the *P. rapae*'s homologue showed no detectable Hi activity.

### Species-specific Hi activity determines the magnitude of *E*-2-GLV emissions from wounded host plants

Hi activity has been confirmed in the OS of numerous lepidopteran species through ex vivo approaches[23,29], and here, we further identify active Hi homologues. However, only *M. sexta* has been reported to induce a pronounced increase in *E*-2-GLVs when feeding on host plants, thereby influencing multitrophic interactions[24,25]. To explore this discrepancy, we applied equal amounts of each species' recombinant Hi protein to wounded leaves of their respective host plants and measured the emission of GLVs. Treatment with MsHi-1 on wounded tomato leaves led to a significant rise in E2AL emissions, compared with water or heat-inactivated MsHi-1 controls (Fig. 3a and Supplementary Fig. 4a). Furthermore, a notable increase in *E*-2-hexenol (E2OL)—a reduction product of E2AL by the plant cinnamaldehyde and hexenal reductase (Fig. 3e)—was also observed (Fig. 3a' and Supplementary Fig. 4a'). By contrast, treatment with CvHi-1 on wounded tomato leaves induced only modest increases in both E2AL and E2OL (Fig. 3b,b' and Supplementary Fig. 4b,b'). Similarly, DpHi-1 on wounded milkweed leaves and BmHi-1 on wounded mulberry leaves elicited only slight increases in E2AL; E2OL was undetectable in these samples (Fig. 3c,d and Supplementary Fig. 4c,d). Taken together, these results indicate that while Hi homologues are widespread in Lepidoptera, MsHi-1 in *M. sexta* shows comparatively high efficiency in converting Z3AL into E2AL under the tested conditions, which in turn led to a pronounced increase in *E*-2-GLVs in our in planta assays.

### Lepidopteran (3*Z*):(2*E*)-hexenal isomerase activity is FAD-dependent

GMC oxidoreductases typically catalyse redox reactions using FAD as a cofactor[30]. Although the conversion of Z3AL to E2AL represents a *cis–trans* rearrangement without a net redox change, we hypothesized that

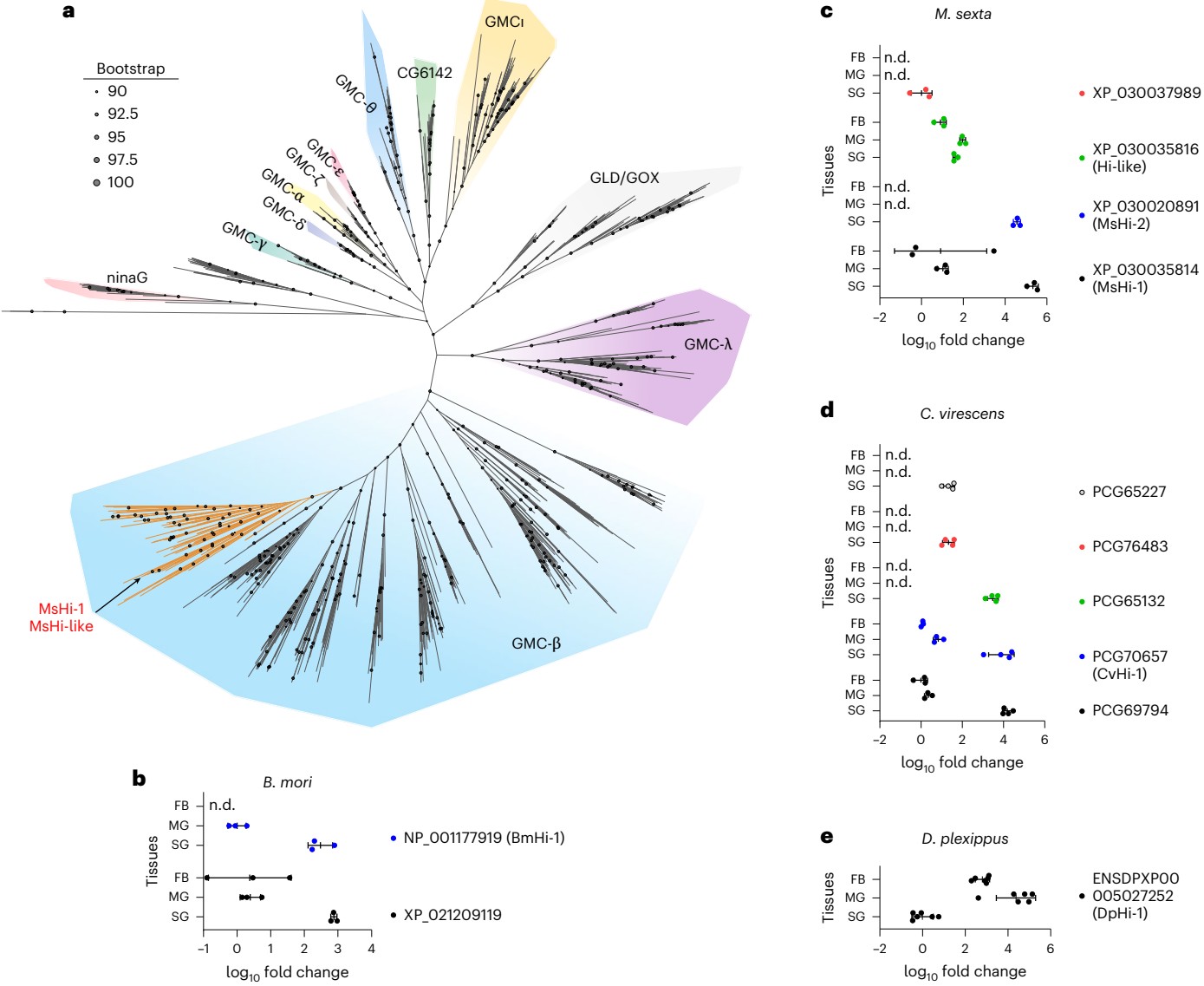

**Fig. 1 | Phylogeny of lepidopteran GMC oxidoreductase proteins and tissue-specific expression profiles of putative Hi genes. a**, A phylogenetic tree was inferred from the alignment of 1251 protein sequences belonging to the GMC oxidoreductase family from 34 lepidopteran species, with three fungal GMC genes considered as outgroup. To assess the monophyly of the Lepidoptera GMC genes within their respective subfamilies, sequences from three Trichoptera species were included. Sequences from *Drosophila melanogaster* were included to serve as reference sequences for accurate classification of GMC subfamilies. The well-supported subclade (bootstrap of 100) within the GMC-β subfamily, referred to as putative Hi clade, which includes two previously reported genes from *M. sexta*, Hi-1 (with Hi activity) and its ortholog Hi-like (without Hi activity),

is highlighted in orange. The tree was analysed by using the IQ-TREE maximum likelihood model. See Supplementary Fig. 1 for the detailed tree with accession numbers and taxon. GLD, glucose dehydrogenase; GOX, glucose oxidase. **b–e**, The gene expression of putative Hi genes, which are clustered in a monophyletic group (bootstrap value of 84) within putative Hi clade (Supplementary Fig. 2), was measured in the fat body (FB), midgut (MG) and salivary glands (SG) of *B. mori* (**b**), *M. sexta* (**c**), *C. virescens* (**d**) and *D. plexippus* (**e**). Expression values were normalized relative to the tissue with the lowest expression level within the species. Genes with Hi activity, based on the results from Fig. 2, are referred to as Hi next to the accession number. *n* = 3 for **b–d**, and *n* = 6 for **e** biologically independent samples. n.d., not detectable.

lepidopteran Hi still requires FAD for its isomerase activity, because the FAD-binding site constitutes a conserved catalytic feature of GMC oxidoreductases. To test this, we measured Hi activity in the presence or absence of supplemental FAD. Control reactions showed that FAD alone did not affect the non-enzymatic conversion of Z3AL to E2AL (Fig. 4a,a′). However, the addition of FAD to enzymatic reactions moderately increased Hi activity across all tested homologues (Fig. 4b–e), as indicated by higher E2AL peak areas in the chromatograms (Fig. 4b′–e′).

Since Hi activity was still detectable even without the addition of extra FAD, we inferred that FAD was already bound to the enzyme during expression in *E. coli* and remained associated through the purification

process. To further validate the importance of FAD in Hi activity, we generated site-directed mutants at key residues predicted to be involved in FAD binding or catalytic activity. Using AlphaFold 3[31] to model MsHi-1 with docked FAD, we identified putative FAD-binding residues. We then compared the predicted structure of MsHi-1 with two experimentally determined GMC oxidoreductase structures: aryl-alcohol oxidase from *Pleurotus eryngii* (PDB: 3FIM)[32] and glucose dehydrogenase from *Aspergillus flavus* (PDB: 4YNT)[33] (Fig. 4f,g). Superimposing their FAD-binding domains onto the predicted FAD-binding domain of MsHi-1 revealed high structural similarity, with root mean square deviation (r.m.s.d.) of 0.22 Å and 0.48 Å, respectively (Fig. 4f′,g′). Y92 in 3FIM and G94 in 4YNT, previously reported as essential residues for FAD binding[32,33]

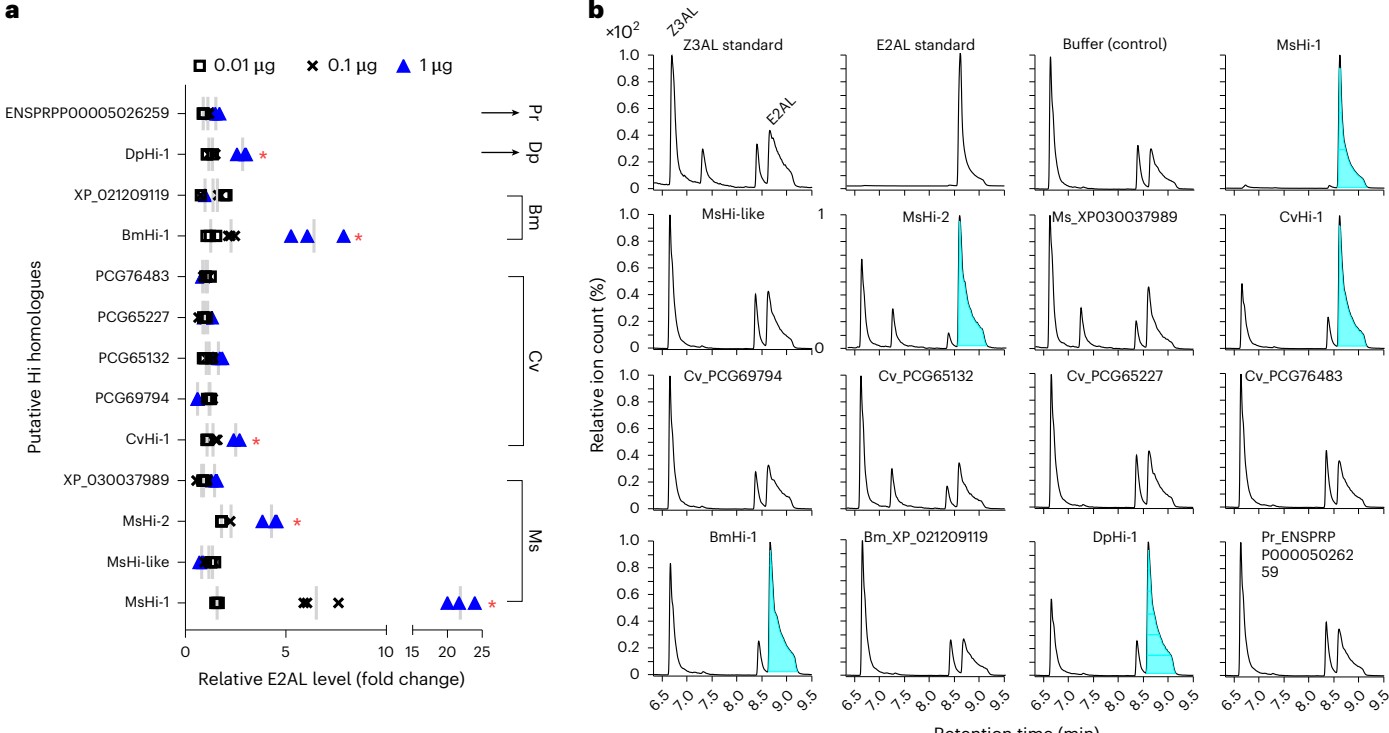

**Fig. 2 | SPME–GC–MS in vitro assay for measuring (3*Z*):(2*E*)-hexenal isomerase activity using purified recombinant proteins. a**, Three different quantities (0.01, 0.1 or 1 µg) of purified recombinant putative Hi proteins were incubated with Z3AL (0.2 mM), and the proportion of E2AL emitted from total aldehydes (Z3AL + E2AL) was calculated. Each protein was tested at the same three concentrations to assess concentration-dependent activity. The buffer control was used to estimate the non-enzymatic conversion rate of Z3AL to E2AL. The values were normalized to the buffer control and are shown as relative fold-change levels. Kruskal–Wallis non-parametric test was performed to assess significant differences between the buffer control and the three different quantities of recombinant proteins. Post hoc pairwise comparison was conducted using the Benjamini–Krieger–Yekutieli two-stage linear step-up procedure. False discovery rate-adjusted *P* values (*q* values) were: MsHi-1 (0.004), MsHi-like (0.22), MsHi-2 (0.005), XP_030037989 (0.17), CvHi-1 (0.02), PCG69794 (0.40), PCG65132 (0.12), PCG65227 (0.06), PCG76483 (0.24), BmHi-1 (0.02), XP_021209119 (0.63), DpHi-1 (0.015) and ENSPRPP00005026259 (0.17). A significant increase in E2AL with increasing recombinant protein quantity is marked with a red asterisk (*q* < 0.05). *n* = 3 biologically independent samples. The grey line indicates median value. Ms, *M. sexta*. Cv, *C. virescens*. Bm, *B. mori*. Dp, *D. plexippus*, Pr, *P. rapae*. **b**, Representative extracted ion chromatograms (ion 55) from the SPME-guided assay with 1 µg of purified protein. The blue highlight on the E2AL peak area indicates a distinct conversion of Z3AL to E2AL. Standard compounds of Z3AL (20 nmol) and E2AL (5 nmol) were used to identify peak areas.

were found to correspond to a conserved histidine (H135) in MsHi-1 (Fig. 4f′,g′) and in all four other lepidopteran Hi homologues that exhibit Hi activity (Fig. 2 and Supplementary Fig. 5). This histidine could potentially form a hydrogen bond with the O4 atom on the isoalloxazine ring of FAD, similar to Y92 and G94 in 3FIM and 4YNT (Fig. 4f′,g′), respectively. Moreover, comparative analysis of conserved C-terminal GMC oxidoreductase active-site motifs[34] showed MsHi-1 and other lepidopteran Hi generally contain H–N pairs (for example, H521/N559 in MsHi-1) (Fig. 4h and Supplementary Fig. 5), where H–H pairs occur in 3FIM (H502/H546) and 4YNT (H505/H548). These two conserved residues are known to be crucial for substrate positioning and electron transfer in GMC oxidoreductases[35,36].

We further introduced alanine substitutions at H135, H521 and N559 to confirm the role of these FAD-interactive residues in Hi activity (Supplementary Fig. 6). The H135A and N559A mutants completely lost Hi activity, even when increasing the amount of protein five times (0.5 µg) (Fig. 4i). The H521A mutant still converted Z3AL to E2AL but at reduced rates. Enzyme kinetics showed an increased $K_m$ (0.04 → 1.01 mM) and a ~25-fold drop in catalytic efficiency (359.6 → 14.0 mM$^{-1}$ s$^{-1}$), whereas $K_{cat}$ remained roughly unchanged (13.8 versus 14.1 s$^{-1}$) (Fig. 4j). In planta assays, using wounded tomato leaves, confirmed these findings (Supplementary Fig. 7). Mechanically wounded leaves that were treated with recombinant protein of either MsHi$_{H135A}$ or MsHi$_{N559A}$ emitted predominantly Z3AL, resembling water-treated controls. MsHi$_{H521A}$-treated leaves released more E2AL than those treated with the other two mutants but still significantly less than wild-type MsHi-1-treated leaves. These findings support the conclusion that the cofactor FAD is essential for lepidopteran Hi activity.

## Distribution and evolutionary history of (3*Z*):(2*E*)-hexenal isomerase in plants and lepidopterans

Previous studies have shown that in plants, an enzyme of the cupin superfamily catalyses the conversion of Z3AL to E2AL[20,21]. Both plant- and lepidopteran-derived Hi display promiscuous activity towards other *Z*-3-aldehydes in vitro, such as *Z*-3-octenal and *Z*-3-nonenal[21,23]. This functional convergence in plants and insects raises intriguing questions about when and how Hi evolved independently in these distinct lineages.

In plants, the cupin superfamily comprises structurally conserved proteins characterized by a β-barrel fold, with each member containing either one or two cupin domains[37]. We first searched for proteins containing the cupin domain (Pfam: PF07883) across 183 representative Viridiplantae (green lineage) species (Supplementary Table 2a). This search yielded 3,734 sequences, which we used to construct a phylogenetic tree, alongside 31 reference sequences from well-characterized cupin superfamily members, including Germin, Vicilin, Legumin, Globulin and plant HI proteins (Supplementary Table 2b). The previously characterized plant HI proteins clustered within a distinct subclade of the

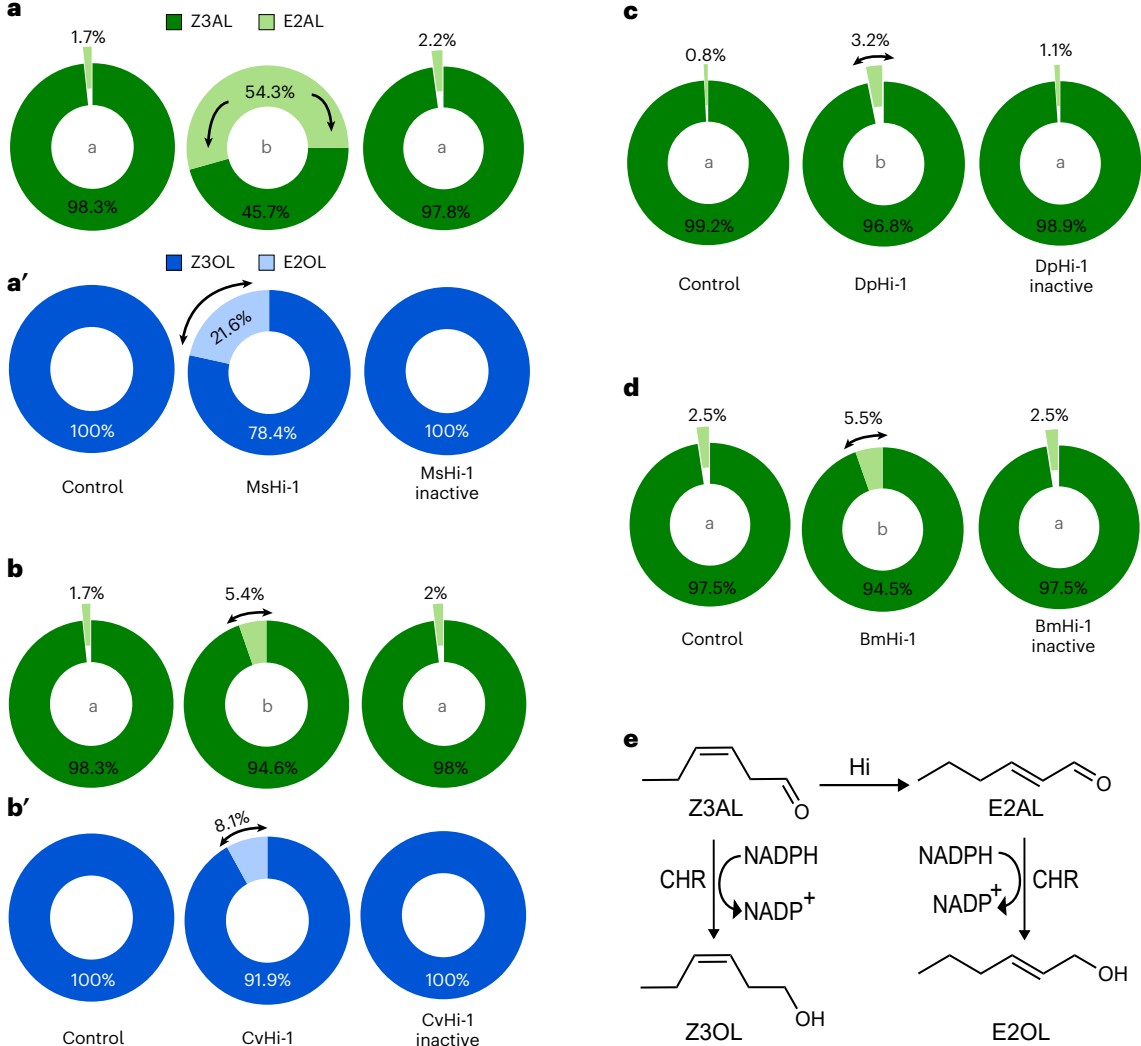

**Fig. 3 | Changes in the emission of GLVs from leaf discs of host plants following treatment with recombinant Hi proteins. a**–**d**, Leaf discs (2.4 mm diameter) from the corresponding host plants were mechanically wounded and treated with 10 µl of Milli-Q water (control), recombinant Hi protein (1 µg) or heat inactive recombinant Hi protein (1 µg). Panels represent treatments with different recombinant Hi proteins: MsHi-1 on tomato (**a**); CvHi-1 on tomato (**b**); DpHi-1 on milkweed (**c**); BmHi-1 on white mulberry (**d**). **a′** and **b′** show the corresponding alcohol products detected in tomato after MsHi-1 and CvHi-1 treatments; alcohols were not detected in milkweed and white mulberry (**c**,**d**). The composition of the $Z3/E2$ form of aldehydes and alcohols was determined using SPME–GC–qToF-MS. A log-normal ordinary one-way ANOVA test was performed to assess significant differences between treatments: **a**: $\eta^2 = 0.98$, $F_{2,12} = 240$, $P < 0.0001$; **b**: $\eta^2 = 0.93$, $F_{2,13} = 87, 26$, $P < 0.0001$; **c**: $\eta^2 = 0.65$, $F_{2,13} = 12, 19$, $P = 0.001$; **d**: $\eta^2 = 0.57$, $F_{2,13} = 8,477$, $P = 0.004$. Different letters in the centre of each pie chart indicate significant differences ($P < 0.05$) by Tukey post hoc test. $n = 5$–6 biologically independent samples. The corresponding dot plot is shown in supplementary Fig. 4. **e**, An llustration of a section of GLV biosynthesis pathway. The conversion of Z3AL to E2AL can occur either spontaneously or through the catalytic action of (3$Z$):(2$E$)-Hi. Both aldehydes can be further reduced to their corresponding alcohols by cinnamaldehyde and hexenal reductase (CHR). Z3OL, $Z$-3-hexenol.

cupin superfamily, which we refer to as the 'HI-potential clade' (Supplementary Fig. 8). Furthermore, our analysis suggests that the proteins in the HI-potential clade first emerged in embryophyte (land plant) hornworts were subsequently lost and later retained in Bryopsida (Supplementary Fig. 9 and Supplementary Table 2c).

Subsequently, a rooted phylogenetic tree was constructed focusing specifically on the sequences within the 'HI-potential clade' (260 sequences) (Supplementary Table 2c). To elucidate their evolutionary relationships (Fig. 5a), we incorporated reference sequences from well-characterized cupin subfamilies—Germin, Vicilin, Globulin and Legumin—that lie outside the HI-potential clade and serve as outgroups. The rooted maximum-likelihood tree grouped the proteins in the Hi-potential clade into seven clades (Fig. 5a). The four clades located near the root of the tree (clade I to IV) include all identified orthologs from ferns, spikemosses, liverworts and hornworts. The remaining sequences formed three distinct crown clades: clade V contains all gymnosperm orthologs, clade VI includes angiosperm orthologs and the clade HI, located at the top of the tree, contains orthologs from mesangiosperms (core angiosperms).

Our data indicate that only proteins in the pink-shaded 'clade HI' possess the critical catalytic residues (H/K/Y) required for Hi activity[20] (Fig. 5b). Moreover, all proteins in clade HI are derived from mesangiosperms, whereas proteins from basal angiosperms (*Amborella trichopoda* and *Nymphaea colorata*, ANA grade) form a separate sister clade (clade VI) that lacks these catalytic residues (Fig. 5a, blue highlight). Interestingly, among 16 monocot species examined, only rice (*Oryza sativa*) possessed proteins within clade HI (Fig. 5a). Hi proteins were entirely absent in 6 of the 16 eudicot families analysed—including Brassicaceae and Caricaceae from the Brassicales order—across the 31 species examined (Fig. 5c).

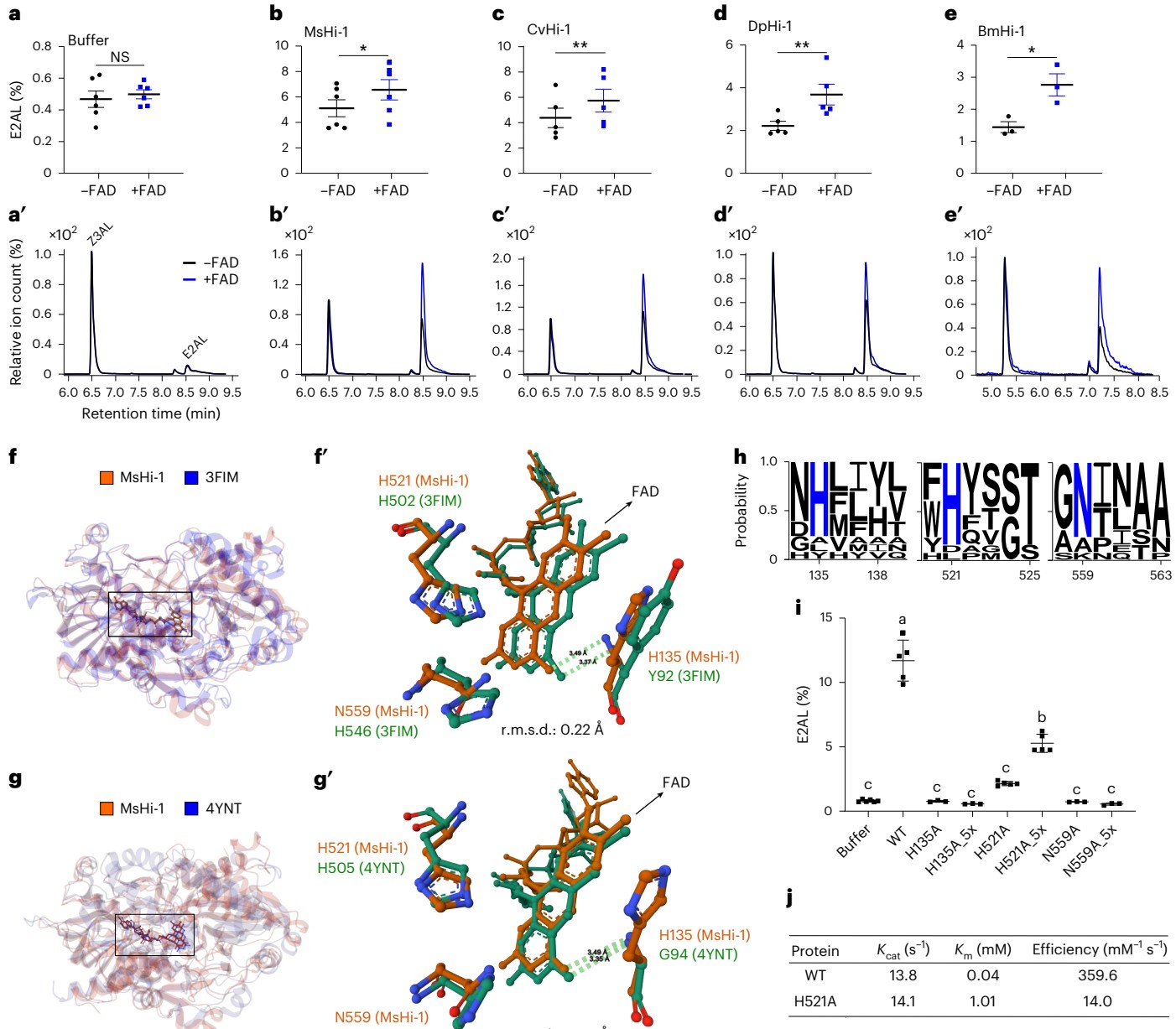

**Fig. 4 | FAD-dependent rearrangement of Z3AL to E2AL by lepidopteran Hi.**
**a–e,** Increased conversion rate of E2AL with FAD addition. Purified recombinant Hi proteins−buffer (**a**), MsHi-1 (**b**), CvHi-1 (**c**), DpHi-1 (**d**) and BmHi-1 (**e**)−were incubated with Z3AL (0.2 mM) with or without FAD (1 mM). The proportion of E2AL emitted from total aldehydes (Z3AL + E2AL) was calculated. Statistically significant differences between treatments were assessed using ratio paired two-tailed $t$-tests: **a**: $\eta_p^2 = 0.17$, $n = 6$ biologically independent samples, $t = 1.01$, $P = 0.36$; **b**: $\eta_p^2 = 0.72$, $n = 6$, $t = 3.55$, $P = 0.017$; **c**: $\eta_p^2 = 0.93$, $n = 5$, $t = 7.16$, $P = 0.002$; **d**: $\eta_p^2 = 0.90$, $n = 5$, $t = 5.94$, $P = 0.004$; **e**: $\eta_p^2 = 0.97$, $n = 3$, $t = 8.66$, $P = 0.013$. The error bars are presented as mean values ± s.d. **a′–e′,** Representative extracted ion chromatograms (ion 69) from SPME−GC−MS for buffer (**a′**), MsHi-1 (**b′**), CvHi-1 (**c′**), DpHi-1 (**d′**) and BmHi-1 (**e′**). **f,g,** Superposition of the MsHi-1 structure with FAD docking (predicted using AlphaFold 3) with structures of aryl-alcohol oxidase from *P. eryngii* (PDB: 3FIM) (**f**) and glucose dehydrogenase from *A. flavus* (PDB: 4YNT) (**g**). **f′,g′,** Detailed view of the FAD-binding pocket shows an N-terminal residue that binds FAD and two conserved C-terminal catalytic residues of GMC oxidoreductases. These N-terminal residues H135 (MsHI-1), Y92 (3FIM) and G94 (4YNT) form a hydrogen bond (green dashed line) with the O4 atom of the FAD isoalloxazine ring. **f′** and **g′** show structural superimposition of the FAD-binding domain of MsHi-1 with homologous structures: 3FIM (**f′**) and 4YNT (**g′**). Structural alignment and r.m.s.d. calculations were performed using Mol* (https://molstar.org/). **h,** Sequence logos representing the conservation of catalytic residues among lepidopteran Hi homologues, and the positions correspond to MsHi-1. Mutated residues are highlighted in blue. Sequence logos were generated using WebLogo 3. **i,** Proportion of E2AL emitted from total aldehydes after incubation of Z3AL with either wild-type (WT) or mutant recombinant MsHi-1 (0.1 µg or 0.5 µg, as indicated by 5×). Significant differences between the buffer control, WT and mutant proteins were determined using one-way ANOVA ($\eta^2 = 0.98$, $F_{7,25} = 144.3$, $P < 0.0001$), followed by Tukey's honestly significant difference post hoc test. Different letters indicate significant differences ($P < 0.05$). $n = 6$ biologically independent samples for buffer control; $n = 5$ for WT and H521A; $n = 3$ for H135A and H559A. The error bars are presented as mean ± s.d. **j,** Kinetic parameters of wild-type and H521A mutant MsHi-1. Catalytic constant ($K_{cat}$) refers to the number of substrate molecules converted to product per enzyme active site per unit time under saturating substrate conditions, while Michaelis constant ($K_m$) represents the substrate concentration at which the reaction velocity reaches half of its maximum value. NS, not significant.

In parallel, we expanded our analysis of the lepidopteran GMC-β oxidoreductase subclade, enriched in Hi homologues (Fig. 1a). Hi homologues were absent in all non-Ditrysian superfamilies—Micropterigoidea, Hepialoidea, Tischerioidea and Palaephatoidea—collectively representing the early diverging lineages of Lepidoptera, as well as in the basal Ditrysian superfamilies Tineoidea and Yponomeutoidea (Fig. 5d and Supplementary Table 1a). Instead, Hi homologues were exclusively identified within the Apoditrysia lineages, including Obtectomera and Macroheterocera. This pattern suggests that Hi arose relatively recently in the lepidopteran lineage, potentially coinciding with the diversification of Apoditrysia. Given that Apoditrysia is estimated to have a crown-group origin in the Early Cretaceous (~118.5 million years ago (Ma))[38], a period marked by rapid angiosperm radiation, it is plausible that the emergence of lepidopteran Hi reflects an adaptive response to increasingly novel and chemically diverse host plants.

## Discussion

In this study, we provide new insights into the evolutionary history and taxonomic distribution of plant and insect Hi genes and describe the enzymatic versatility of Hi genes in Lepidoptera. Functional assays of lepidopteran Hi homologues revealed that at least one gene of each examined species exhibited measurable Hi activity in vitro. Notably, Hi-1 from *M. sexta* induced significantly higher levels of E2AL and downstream E-2-GLVs in host plants compared with Hi from other three lepidopteran species (Fig. 3a). These results, obtained in situ by applying recombinant Hi proteins onto mechanically wounded leaves of host plants, support previous observations that plants fed on by *M. sexta* release significantly higher levels of E2AL compared with plants fed on by *C. virescens*[26]. Given that the previous study demonstrated that Hi also plays an essential developmental role independent of Z3AL isomerization this indicates that in some Lepidoptera, Hi may primarily serve physiological rather than ecological functions[23].

The ecological relevance of Hi is well established in the *M. sexta* system: shifts in the ratio of *Z*-3- to *E*-2-GLVs, driven by Hi activity, serve as cues for both ovipositing moths and natural enemies of the herbivore[24,25]. However, whether Hi ultimately benefits the insect or the plant remains unclear, as its effects can be both advantageous and detrimental depending on context. Similarly, other OS effectors, such as fatty acid dehydratases, hexenal-trapping molecules and glucose oxidase, can reduce GLV emissions[39–41], which may dampen the strength of volatile cues in the environment. Moreover, suppression of GLVs via fatty acid dehydratases has been shown to decrease parasitoid attraction, potentially benefiting the herbivore by reducing attack risk[40]. Beyond its ecological effects, we previously demonstrated that Hi enzymes act on a range of *Z*-3-alkenal substrates and that *hi* mutants of *M. sexta* raised on artificial diet lacking GLVs developed more slowly and showed increased rates of adult abnormalities compared with wild-type insects[23]. This suggests that Hi may also serve internal physiological roles independent of interactions with GLV-producing plants. Together, these findings highlight that variation in Hi activity among lepidopteran species probably reflects a combination of ecological, physiological and evolutionary factors.

Homology analysis indicates that lepidopteran Hi proteins belong to the GMC oxidoreductase protein family. Given its involvement in an electrophilic isomerization reaction, this enzyme could be classified as an intramolecular oxidoreductase (EC 5.3). A notable example of such an intramolecular oxidoreductase is isopentenyl pyrophosphate isomerase from plants, which participates in isoprenoid biosynthesis. This enzyme catalyses the isomerization of isopentenyl pyrophosphate into its electrophilic isomer, dimethylallyl diphosphate (DMAPP), via a protonation/deprotonation mechanism[42,43]. Our results demonstrated that FAD is essential for lepidopteran Hi activity, even though there is no net redox change between the substrate and the product (Fig. 4). Although the precise role of FAD in Hi catalysis remains unresolved, one possibility is that it functions analogously to polyunsaturated fatty acid isomerase (PAI) from *Propionibacterium acnes*[44,45]. In PAI, FAD facilitates the double bond isomerization of linoleic acid to conjugated linoleic acid through an ionic mechanism. During this reaction, FAD stabilizes a carbocation intermediate by interacting with the substrate's double bonds via its redox-active isoalloxazine ring. Another example of a FAD-dependent non-redox reaction is carotene *cis*–*trans* isomerase (CRTISO) in plants, where FAD facilitates the isomerization of prolycopene to all-*trans*-lycopene during carotenoid biosynthesis[46,47]. These examples suggest that FAD in lepidopteran Hi may facilitate a coupled isomerization and double bond migration from *Z*-3- to *E*-2-aldehydes by stabilizing the transient enolate intermediate and aiding proton abstraction.

The Z3AL isomerization by plant HI has previously been proposed to involve a keto-enol tautomerism mechanism[20]. The process is mediated by a conserved catalytic histidine (Fig. 5b), which could abstract a proton from the C2 position of Z3AL, forming a transient enolate intermediate. This enolate structure may allow the electron density to shift, facilitating the formation of a keto-like tautomer. We identified an N-terminal histidine in Lepidoptera Hi (Fig. 4h and Supplementary Fig. 5), which is critical for Hi activity. As this conserved histidine has been previously reported in other GMC oxidoreductases to abstract the proton from the substrate[33,34], this histidine may perform a role that is similar to a catalytic histidine in plant HI. However, direct biochemical evidence confirming its involvement in the Z3AL isomerization remains to be established. Some of the inactive Hi homologues show substitution at catalytic residues such as leucine and aspartic acid replacing the N-terminal histidine in PCG65132 and PCG76483 of C. *virescens*, or alanine, leucine or tyrosine replacing the C-terminal histidine critical for FAD binding in PCG69794 and PCG76483 of C. *virescens* and the homologue of *P. rapae*, respectively (Supplementary Fig. 5). Nevertheless, several lepidopteran Hi proteins with all three conserved FAD-binding and catalytic sites are still unable to rearrange Z3AL to E2AL (Fig. 2 and Supplementary Fig. 5). This includes the gene duplicate of MsHi-1, MsHi-like (Fig. 2), which shares 85% coding sequence identity. A similar observation was made in a plant

**Fig. 5 | Divergence time of Hi proteins in plants and lepidopterans.**
**a**, A maximum likelihood phylogenetic tree was inferred from the protein sequences that clustered within the HI-potential branch of the cupin superfamily tree (highlighted in Supplementary Fig. 8), alongside representative cupin superfamily proteins from outside this branch (labelled as Germin, Vicilin, Legumin and Globulin). The clade HI (highlighted in red) includes proteins with conserved HI catalytic residues (His/Lys/Tyr). This clade is found exclusively in mesangiosperms, based on evidence that proteins from basal angiosperms (ANA grade), highlighted in blue, lack these catalytic residues. The bootstrap values were inferred from RAxML-NG and IQ-TREE 2. **b**, Sequence logos represent the alignment of proteins from each subclade. The HI catalytic residues (His/Lys/Tyr) in proteins of the HI clade are highlighted in blue. **c**, Distribution survey of HI proteins across eudicot families. Filled circles indicate HI presence in all analysed species within a family, half-filled circles show presence in some species, and empty circles indicate complete absence. All analysed species from both Brassicaceae and Caricaceae families (highlighted in green), members of the Brassicales order, lack HI proteins. Sequence logos were generated using WebLogo 3. **d**, The chronogram illustrates the emergence and distribution of lepidopteran Hi across lineages in relation to major geological events. Hi proteins appeared in the Apoditrysia lineage of Lepidoptera (132.1–105.6 Ma), coinciding with the angiosperm radiation (125–90 Ma). In plants, HI emerged in mesangiosperms (192.2–166.4 Ma). The presence of Hi in lepidopteran species was inferred from proteins that clustered within the putative Hi clade in the phylogenetic tree (Fig. 1a). Presence of Hi in different lepidopteran superfamilies is indicated using filled, half-filled and empty circles, similar to **c**. Estimates of lepidopteran and plant divergence times are based on Kawahara et al.[38], Peris and Condamine[78] and Yang et al.[54].

Hi: one of the *Cucumis sativus* Hi proteins (Cs033080/XP_011651276), which showed no activity[21], despite possessing all catalytic HKY residues and clustering within the clade HI in the phylogenetic tree (Fig. 5a and Supplementary Table 2b). Future studies are needed to compare and test the roles of surrounding residues directly involved in Z3AL

interactions between active and inactive lepidopteran Hi, as well as Hi with varying activity levels. Insights from plant HI suggest that a tyrosine near the catalytic histidine (Tyr-128 in *C. annuum*) may form the substrate binding pocket, highlighting the importance of surrounding residues in determining HI functionality[20].

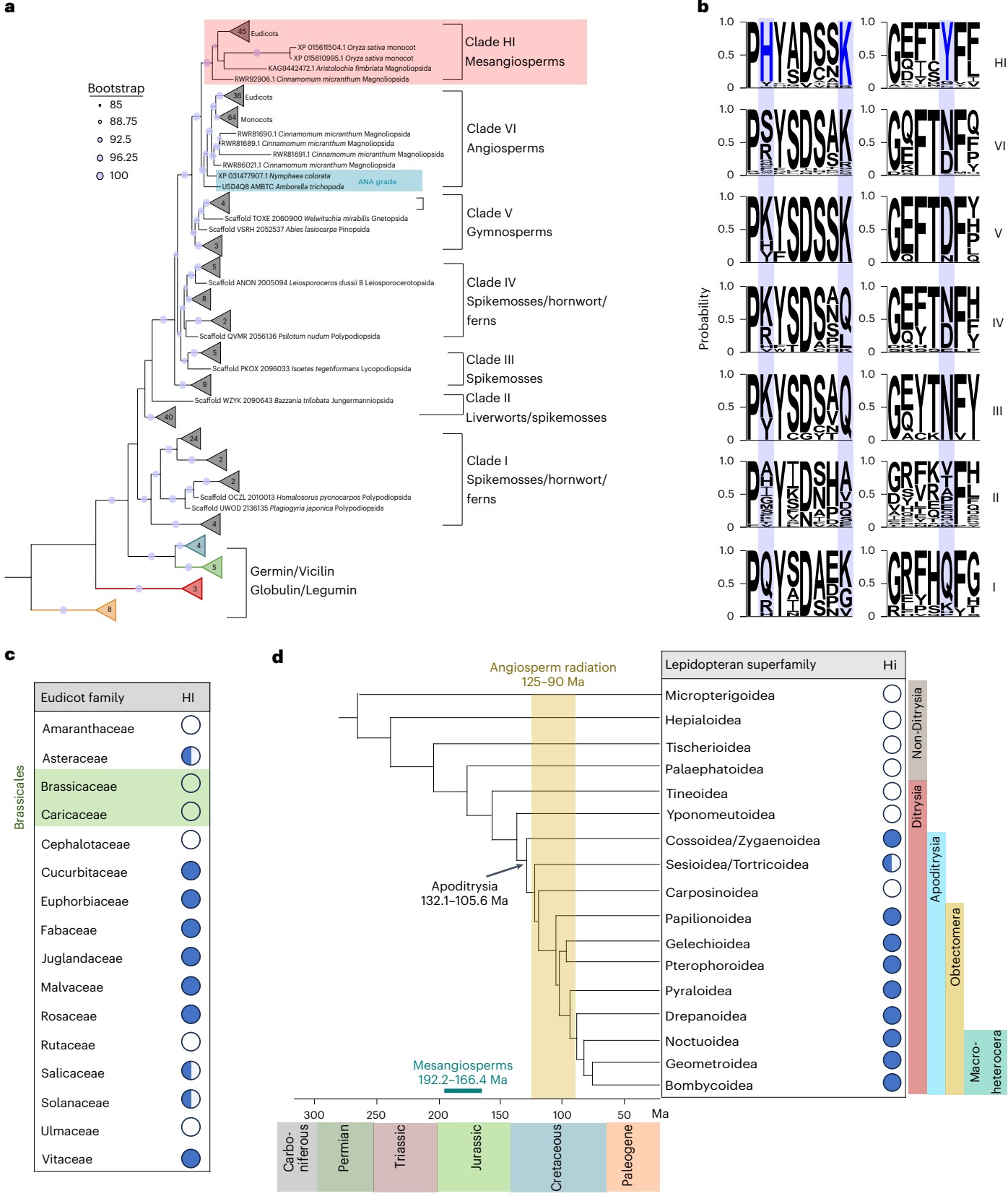

A previous study has shown that mechanical damage in Cucurbitaceae and Fabaceae species leads to the emission of E2AL as the dominant GLV aldehyde, with relative proportions of ~63–87% of the total aldehyde pool[14]. Consistent with this, we found that all seven Cucurbitaceae and Fabaceae species in our phylogenetic analysis possess plant HI homologues containing the catalytic HKY residues (Fig. 5c). By contrast, five Brassicaceae or Caricaceae species within the Brassicales order lack HI proteins (Fig. 5c). This is in agreement with earlier reports that Brassicales species emit Z3AL rather than E2AL upon mechanical damage[29]. Interestingly, two lepidopteran specialists that feed on Brassicales—*P. rapae* and *Plutella xylostella*—also lack a functional Hi. Although *P. rapae* possesses a sole homologue clustering within the putative Hi subclade (Fig. 1a and Supplementary Fig. 2), no activity was detected either from the recombinant protein (Fig. 2a) or from the OS[23]. Meanwhile, *P. xylostella* has no representative gene within the putative Hi clade in the phylogenetic tree (Supplementary Table 1a). One plausible explanation is that the glucosinolate-based defences of Brassicales reduce the ecological importance of Z3AL to E2AL conversion, thereby relaxing selection for Hi in both the plants and their co-evolved herbivores. Although some herbivores, such as *M. sexta*, can rely on GLVs for feeding stimulation[48] or oviposition cues[24], Brassicales specialists often depend primarily on glucosinolates for both feeding stimulation and oviposition decisions[49]. Similarly, although certain natural enemies use shifts in the *Z*-3/*E*-2-GLV ratio for prey detection[25,40,50], parasitoid wasps that target Brassicales-feeding caterpillars (*Cotesia rubecula* and *Hyposoter ebeninus*) instead rely, although not exclusively, on glucosinolate-derived volatiles (for example, isothiocyanates and nitriles) as their host location cues[51–53]. This may have consequently attenuated the selective pressures driving the maintenance of Hi function in both Brassicales and their specialist herbivores. Beyond ecological relaxation, gene-family architecture may also help explain why some lepidopteran species, such as *P. xylostella*, lack an identifiable Hi homologue without apparent fitness loss, whereas *Manduca* Hi-1 mutants show developmental defects. This observation suggests that the developmental role of MsHi-1, in addition to its characterized Hi activity, may be functionally compensated in other lepidopteran lineages by GMC-β homologues devoid of Hi activity. Such compensation is consistent with expansions of the GMC-β subfamily in Lepidoptera, with numerous paralogs implicated in development and immunity[27,28], thereby reducing the selective pressure to retain a canonical *Hi* gene.

Phylogenetic analysis (Fig. 5a) and the generated chronogram (Fig. 5d) indicate that plant HI first arose in the stem lineage of mesangiosperms (192.2–166.4 Ma)[54], with no orthologues detectable in basal angiosperms (*A. trichopoda* and *N. colorata*) (Fig. 5a). This finding implies that the evolution of plant HI postdates the origin of the GLV pathway, as the capability for GLV biosynthesis and the associated key enzyme, hydroperoxide lyase, were already established in early vascular plants, including lycophytes and monilophytes[55]. Lepidopteran Hi independently evolved approximately 60 million years later during the early Cretaceous radiation of Apoditrysia (132.1–105.6 Ma)[38], coinciding with—and probably facilitated by—the diversification of flowering plants during this period. Consistent with this pattern, several adaptive traits in Apoditrysia, such as a versatile proboscis and expanded detoxification gene families, reflect evolutionary adjustments to novel ecological niches created by angiosperm diversification in the Cretaceous[56,57]. This temporal pattern, in which plant metabolic innovations precede and potentially drive insect biochemical adaptations, reflects a broader pattern seen in other evolutionary arms races. A representative example is the glucosinolate detoxification through nitrile-specifier proteins (NSPs): Kelch-type NSPs emerged in Brassicales after a whole-genome duplication event (~85 Ma; At-β event), shortly before insect-specific NSPs independently evolved in Pierinae butterflies (~68 Ma) to redirect host plants' glucosinolates

towards less-toxic nitriles[58–61]. Collectively, the independent emergence of Hi in both mesangiosperms and Lepidoptera represents another example of this evolutionary scenario, potentially enhancing the ecological specificity and impact of GLV-mediated interactions across trophic levels.

## Methods

### Protein nomenclature
The abbreviation of (3*Z*):(2*E*)-hexenal isomerase in plants is written as HI, in accordance with the *Arabidopsis* nomenclature system and previous plant HI studies[20,21]. For describing insect proteins, or when simultaneously describing both insect and plant proteins, we use Hi, in accordance with the *Drosophila* nomenclature system and previous lepidopteran Hi study[23]. When referring to general enzyme activity (for example, Hi activity), we consistently use Hi to maintain a unified terminology across taxa.

### Insects and plants
Rearing conditions of tobacco hornworm (*M. sexta*), tobacco budworm (*C. virescens*), cabbage white (*P. rapae*) and monarch butterfly (*D. plexippus*) are described in previous articles[23,62,63]. For dissecting different tissues, fourth to fifth instar larvae were dissected in 1× PBS buffer (pH 7.4) to extract midgut, fat body and salivary glands. Tissues from three different individuals were pooled for each biological replicate. All samples were flash-frozen in liquid nitrogen and stored at −80 °C. Tomato Micro-Tom (*Solanum lycopersicum*), white mulberry (*Morus alba*) and milkweed (*Asclepias incarnate*) were grown in the greenhouse with a day–night cycle of 16 h (26 –28 °C)–8 h (22 –24 °C) under supplemental light from Master Sun-T PIA Agro 400 or Master Sun-T PIA Plus 600-W sodium lights (Philips).

### RNA extraction and cDNA synthesis
The collected tissues were ground in liquid nitrogen with sterile pestles. Total RNA was extracted using the TRIzol/chloroform method according to the manufacturer's protocol. The purified RNA was treated with DNase using the Ambion Turbo DNase kit (Thermo Fisher Scientific) to remove genomic DNA. Total RNA concentration was measured by NanoDrop ND-1000 (Thermo Fisher Scientific). One microgram of total RNA was used for cDNA synthesis with RevertAid First Strand cDNA synthesis kit (Thermo Fisher Scientific). Quantitative real-time PCR (ABI 7500 Real-Time PCR System; Applied Biosystems) was performed using the HOT FIREPol EvaGreen qPCR Mix Plus (Solis BioDyne).

### Gene cloning and recombinant protein production
The coding regions of putative Hi homologues from *C. virescens*, *M. sexta* and *D. plexippus* were PCR amplified from a cDNA mixture derived from salivary glands and midgut tissues. The primers used for amplification are listed in Supplementary Table 3. The coding sequences of *B. mori* were obtained through de novo synthesis (Gene Universal). All coding sequences were cloned into the pGEX-4T-1 vector for GST-fusion protein expression in *E. coli*. Plasmids were transformed into competent *E. coli* BL21 (DE3) for recombinant protein expression. A single colony of transformed *E. coli* was cultured in 10 ml LB medium, shaking overnight at 37 °C. The overnight *E. coli* culture was then transferred to 1 litre of 2× YT medium (16 g tryptone, 10 g yeast extract and 5 g NaCl) and kept shaking at 37 °C until the optical density at 600 nm reached 0.4–0.5. IPTG (1 mM final concentration) was added to induce recombinant protein expression, and the culture was incubated with shaking at 16 °C for 24–48 h. For MsHi-2, induction was performed with 0.2 mM IPTG at 10 °C for 120 h. The *E. coli* pellet was collected by centrifugation at 15,000*g*. After discarding the supernatant, the pellet was snap-frozen in liquid nitrogen and stored at −20 °C until use. For purification of recombinant proteins, the *E. coli* pellet was first resuspended in 30 ml of lysis buffer containing 1× PBS (pH 7.3), 1 mM EDTA, 10 mg ml$^{-1}$ lysozyme and proteinase inhibitor cocktails

(50 ml per tablet). The suspended pellet was sonicated on ice, followed by the addition of 1% Triton X-100 and rotated for 30 min. The *E. coli* lysate was collected by centrifugation at 15,000*g* and passed through a 0.45 µm filter. The lysate was batch purified using GST Sepharose 4B (GE Healthcare) according to the manufacturer's instructions, and the purified proteins were preserved in 50 mM Tris−HCl buffer (pH 8.0) with 10% glycerol at −80 °C until use. Quantitative densitometry of proteins from SDS−polyacrylamide gel electrophoresis stained with Coomassie Blue was used to determine protein concentrations by comparing the relative intensity of bands between recombinant proteins and a BSA standard (Bio-Rad Image Lab).

## Western blot analysis

Recombinant proteins were mixed with 4× loading buffer and boiled at 95 °C for 3 min. The proteins were separated on a 10% SDS−polyacrylamide gel electrophoresis gel and transferred to an Immobilon-E PVDF membrane (Millipore) by semi-dry blotting. The membrane was washed three times with 1× PBST (0.05% Tween 20) for 15 min each, then blocked with 5% BSA at room temperature for one hour. Subsequently, the membrane was incubated with GST-HRP conjugated antibody (1:2,000, Santa Cruz Biotechnology) on a rotator overnight at 4 °C. After three additional washes with 1× PBST for 15 min each, the membrane was treated with 1 ml of chemiluminescence solution (100 mM Tris−HCl pH 8.5, 9 ml of $H_2O$, 1 ml of 1 M Tris−HCl pH 8.5, 22 µL of 90 mM *p*-coumaric acid, 50 µl of 200 mM luminol and 3 µl of 30% $H_2O_2$). Images were captured using the Odyssey Fc Imaging System (LI-COR) and analysed with Image Studio Lite or the ChemiDoc MP Imaging System and analysed with Bio-Rad Image Lab.

## Homologue identification and phylogenetic analysis in Lepidoptera

A total of 34 lepidopteran species representing a broad range of superfamilies were selected to capture the phylogenetic diversity of the order. In addition, three species from *Trichoptera* (caddisflies), the sister group of *Lepidoptera* and *Drosophila melanogaster* (Diptera) were included to assist in the classification of GMC oxidoreductase subfamilies[27,28]. The proteome sequences of studied organisms were obtained from the NCBI (https://www.ncbi.nlm.nih.gov/) and Ensembl (https://beta.ensembl.org/) databases (Supplementary Table 1a). Genome assemblies were prioritized based on completeness and the availability of annotated protein-coding genes. For lineages lacking annotated genomes, specifically Hepialoidea, Tischerioidea and Palaephatoidea, transcriptome shotgun assemblies were used as alternative sources of protein-coding sequences. From these transcriptome assemblies, open reading frames were predicted using the getorf program in the EMBOSS suite (version 6.5.7)[64].

To identify GMC oxidoreductase homologues, we employed a domain-based search strategy using profile HMMs. HMM profiles for the N-terminal (PF00732) and C-terminal (PF05199) domains of the GMC oxidoreductase family were downloaded from the Pfam database hosted by InterPro (https://www.ebi.ac.uk/interpro/). These profiles were combined into a single database and indexed using hmmpress from the HMMER suite (v3.3.2)[65]. These validated sequences were then extracted from the original protein FASTA files, yielding a refined set of full-length GMC homologues. To reduce redundancy from alternative splicing, isoforms were collapsed by retaining only the longest protein variant per gene. Three fungal GMC sequences (AAF59929.2, XP_001727544.1 and AAD01493.1) were incorporated as outgroup taxa. The final dataset comprised 1,251 curated GMC domain sequences used for phylogenetic analysis. Species names, taxonomic classifications and sequence sources are detailed in Supplementary Table 1b.

For phylogenetic inference, protein sequences were aligned using MAFFT (v7.520)[66] with the E-INS-i algorithm. The resulting multiple sequence alignment was trimmed with ClipKIT (v2.0.1)[67]. The best-fitting amino acid substitution model was selected using ModelTest-NG (v0.1.6)[68]. Maximum likelihood phylogenetic analysis was conducted with IQ-TREE 2 (v2.3.4)[69]. Branch support was assessed with 3,000 ultrafast bootstrap replicates and 3,000 SH-aLRT tests. The final phylogenetic tree was rooted with the predefined fungal outgroup sequences.

## Homologue identification and phylogenetic analysis in the green lineage (Viridiplantae)

The complete predicted proteome sequences (Supplementary Table 2a) were obtained from the NCBI GeneBank (https://ncbi.nlm.nih.gov), UniProt-Proteomes database (https://www.uniprot.org) and JGI (http://genome.jgi.doe.gov). The 1000 Plants project (OneKP) database (http://www.onekp.com) was an additional source for predicted proteome sequences inferred from transcriptomic data. All cupin-domain(PF07883)-containing proteins were identified on the previously retrieved predicted proteome sequences using the HMM-based tool hmmsearch[70]. All identified sequences (3,734 sequences) were combined with 31 representative sequences from cupin protein subfamilies (including Germin, Vicilin, Legumin, Globulin and Hi) (Supplementary Table 2b) into one single dataset and analysed using a phylogenetic approach (Supplementary Fig. 8). Only sequences that clustered within the HI-potential clade in the phylogeny were identified as potential HI orthologs (Supplementary Table 2c). SMART and Pfam databases were employed to identify conserved domains present in potential HI orthologs[71,72]. The results from both databases were merged, redundant domains were filtered-out and domain architecture was analysed using the HMM-based tool hmmscan[73].

All identified potential HI orthologs and representative sequences from cupin protein subfamilies were aligned using MAFFT and ambiguously aligned regions were removed with trimAl[74]. The resulting alignments were evaluated with ProtTest v3[75] to determine the best-fit model for amino acid substitution. Two separate maximum-likelihood phylogenetic analyses were computed using RAxML-NG[76] and IQ-TREE 2[69] (Fig. 5a), each with 1,000 bootstrap replicates. Bootstrap support values from both analyses were mapped onto the IQ-TREE 2 maximum-likelihood tree, which was rooted at its midpoint (Fig. 5a).

## (3*Z*):(2*E*)-hexenal isomerase (Hi) activity assays by SPME−GC−qToF-MS

A 200 µl solution containing recombinant proteins in a 20 mM Tris−HCl buffer (pH 8.5) was first transferred to a 1.5 ml GC vial equipped with a 200 µl insert. The reaction was initiated by adding Z3AL (0.2 mM final concentration) to the solution. The GC vial was gently vortexed for 1 min. Subsequently, 100 µl mixture was transferred to a 20 ml glass headspace vial (SureSTART, Thermo Scientific) and immediately closed with a cap crimper. Headspace volatiles were collected using the SPME fibre (carboxen/polydimethylsiloxane coated) for 10 min at 35 °C and analysed by GC−qToF-MS. After sampling, the fibre was desorbed for 1 min in the injection port which was constantly kept at 250 °C. Compounds were separated on a HP-5ms column (30 µm × 250 µm, 0.25 µm film thickness; Agilent) in an Agilent 7890A gas chromatograph with a temperature program set to 40 °C for 5 min, increasing to 140 °C at a rate of 5 °C min$^{-1}$, followed by increasing temperature to 250 °C at a rate of 15 °C min$^{-1}$ and an additional 5 min at 250 °C. Helium was used as the carrier gas with the transfer column flow set to 3 ml min$^{-1}$ and a flow rate of 1 ml min$^{-1}$ thereafter. Mass spectra were generated by an Agilent 7200 accurate-mass quadrupole time-of-flight mass spectrometer, operating in electron ionization mode (70 eV) at 230 °C and collected with an acquisition rate of five scans per second. Volatiles were identified and quantified using standard volatiles listed in Supplementary Table 3. The conversion rate of E2AL was calculated as the proportion of its intensity relative to the total aldehyde intensity (Z3AL + E2AL). Low background levels of E2AL were consistently detected in each sample as a result of the spontaneous non-enzymatic conversion of Z3AL during measurements. To correct for this, the

calculated value was adjusted by subtracting the non-enzymatic conversion rate determined from a buffer-only control, yielding the relative E2AL level (Fig. 2a). To account for differences in detector sensitivity, response factors for Z3AL and E2AL were calculated using measured intensities of 2 nmol of each standard compound.

### Effect of additional FAD on Hi activity

To compare the activity of different Hi enzymes with and without additional FAD, the following amounts of recombinant proteins were used: 3 µg BmHi-1, 1.38 µg DpHi-1, 1 µg CvHi-1 and 0.0625 µg MsHi-1. These protein concentrations were carefully selected to ensure that, without FAD, the conversion of Z3AL to E2AL would not reach completion, allowing any potential increase in activity upon FAD addition to be clearly observed. Each enzyme was tested in a 200 µl reaction mixture containing 20 µg BSA, with or without 1 mM FAD, in 20 mM Tris–HCl buffer (pH 7.0). This mixture was transferred to a 1.5 ml GC vial equipped with a 200 µl insert, after which 0.2 mM Z3AL was added to initiate the reaction. The vial was gently vortexed for 1 min. Subsequently, 100 µl of the mixture was transferred to a 20 ml glass headspace vial and sealed immediately using a cap crimper. Headspace volatiles were collected using the SPME fibre for 10 min at 35 °C and analysed by GC–qToF-MS.

### Enzyme kinetics

For the determination of the kinetic parameters of the MsHi wild-type and H521A mutant, a substrate concentration range of 5–4,000 µM Z3AL was used for the MsHi wild-type and 125–4,000 µM Z3AL for the mutant. Recombinant proteins were diluted with 20 mM Tris–HCl (pH 8.5) buffer to a final volume of 200 µl. A total of 37.5 ng of MsHi wild-type and 112.5 ng of H521A mutant were used for measurement. The mixture was incubated at room temperature for 2 min, subsequently transferred to 20 ml glass headspace vial and analysed by GC–qToF-MS. The $K_m$, $K_{cat}$ and enzyme efficiency ($K_{cat}/K_m$) were calculated with nonlinear least-square regression using GraphPad Prism 10.

### Analysis of volatiles in planta

A 24 mm diameter of leaf disc was punched out from the lamina of the second or third pair of true leaves from the top of 4-week-old plants. The leaf disc was mechanically wounded on the adaxial surface by rolling a fabric pattern wheel to produce two parallel rows of punctures on either side of the midvein. A total of 10 µl of recombinant protein, heat-inactivated recombinant protein (boiled at 95 °C for 3 min) or water was applied to the wounds and gently dispersed across the leaf surface. After 20 s, the leaf disc was transferred to a 20 ml glass headspace vial, closed with a crimp cap and volatiles were immediately collected with a SPME fibre for 10 min at 35 °C and measured by GC–qToF-MS analysis.

### Statistical analysis

Statistical analyses were performed using GraphPad Prism 10. The Shapiro–Wilk test was used to assess normality of data distributions. For comparisons among multiple groups, one-way analysis of variance (ANOVA) followed by Tukey's multiple comparisons test was used. For non-parametric data, a Kruskal–Wallis test followed by Dunn's multiple comparisons test was applied. Different letters above bars in the graphs indicate statistically significant differences between groups ($P < 0.05$). Two-tailed $t$-tests were used for comparisons between two groups. The error bars in Figs. 2a and 4b–e represent mean values ± s.d. (*$P < 0.05$, **$P < 0.01$, ***$P < 0.001$).

### Reporting summary

Further information on research design is available in the Nature Portfolio Reporting Summary linked to this article.

### Data availability

All reagents are available in Supplementary Table 3. The data that support the findings of this study are available in the Supplementary Information

and via Zenodo at https://doi.org/10.5281/zenodo.17853039 (ref. 77). Source data are provided with this paper.

### Code availability

The analysis for the plant hexenal isomerase is available via GitHub at https://github.com/abdo3a/Hexenal_Isomerases_Viridiplantae. The analysis for the lepidopteran hexenal isomerase is available via GitHub at https://github.com/bulahwoo/Hexenal_Isomerases_Lepidoptera.

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

## Acknowledgements

We thank E. Poelman (Wageningen University) who kindly provided us with *P. rapae*. We appreciate the help of our colleagues at University of Amsterdam, L. Tikovsky and H. Lemereis for taking care of all plants in the glasshouse, J. Vreede and S. Eltschkner for the discussion of protein structure analysis, A. Groot provided us with *C. virescens*. We thank H. Vogel (Max Planck Institute), D. Doležel (Biology Centre CAS) and S. Vlastimil (Biology Centre CAS) for valuable discussion on this work. Computational resources for homologue identification and phylogenetic analysis in Lepidoptera were provided by the e-INFRA CZ project (ID: 90254), supported by the Ministry of Education, Youth and Sports of the Czech Republic and the ELIXIR-CZ project (ID: 90255), part of the international ELIXIR infrastructure. A.S. was supported by the SequAna Sequencing Analysis Core Facility at the Department of Biology, University of Konstanz. C.-W.T., R.J.S. and J.G.A. were supported by US National Science Foundation (NSF-IOS 1754996). This work was supported by the European Research Council (ERC) under the European Union's Horizon 2020 research and innovation programme (grant agreement no. 805074) (S.A. and Y.-H.L.).

## Author contributions

Y.-H.L. conceived and designed the study, performed experiments, analysed data and drafted the paper. B.C.h.W. and A.S. analysed the phylogeny. S.M.E.H. performed cloning and quantitative PCR for *D. plexippus*, *C. virescens* and MsHi-1. I.P. performed GC/MS analysis and kinetic measurements of mutant Hi. C.-W.T., J.G.A. and R.J.S. conducted quantitative analysis for *D. plexippus* Hi. S.A. conceived the study, supervised the project, acquired funding and drafted the paper. All authors read and approved the final paper.

## Competing interests

The authors declare no competing interests.

## Additional information

**Correspondence and requests for materials** should be addressed to Yu-Hsien Lin or Silke Allmann.

# Reporting Summary

## Statistics

For all statistical analyses, confirm that the following items are present in the figure legend, table legend, main text, or Methods section.

| n/a | Confirmed | |
|---|---|---|
| ☐ | ☒ | The exact sample size (*n*) for each experimental group/condition, given as a discrete number and unit of measurement |
| ☐ | ☒ | A statement on whether measurements were taken from distinct samples or whether the same sample was measured repeatedly |
| ☐ | ☒ | The statistical test(s) used AND whether they are one- or two-sided *Only common tests should be described solely by name; describe more complex techniques in the Methods section.* |
| ☒ | ☐ | A description of all covariates tested |
| ☐ | ☒ | A description of any assumptions or corrections, such as tests of normality and adjustment for multiple comparisons |
| ☐ | ☒ | A full description of the statistical parameters including central tendency (e.g. means) or other basic estimates (e.g. regression coefficient) AND variation (e.g. standard deviation) or associated estimates of uncertainty (e.g. confidence intervals) |
| ☐ | ☒ | For null hypothesis testing, the test statistic (e.g. *F*, *t*, *r*) with confidence intervals, effect sizes, degrees of freedom and *P* value noted *Give P values as exact values whenever suitable.* |
| ☒ | ☐ | For Bayesian analysis, information on the choice of priors and Markov chain Monte Carlo settings |
| ☒ | ☐ | For hierarchical and complex designs, identification of the appropriate level for tests and full reporting of outcomes |
| ☐ | ☒ | Estimates of effect sizes (e.g. Cohen's *d*, Pearson's *r*), indicating how they were calculated |

*Our web collection on statistics for biologists contains articles on many of the points above.*

## Software and code

Policy information about availability of computer code

| Data collection | Applied Biosystems 7500 Real-Time PCR SDS software was used to collect qPCR result. Agilent MassHunter Quantitative Analysis was used to collect GC-MS results. Odyssey® Fc Imaging System (LI-COR) and Image Studio Lite (ver 5.2) were used to visualize western blotting result. |
|---|---|
| Data analysis | Image J was used to quantify the ban intensity on SDS-PAGE gel. Microsoft Excel and Graphpad Prism 10 were used for statistical analysis. |

For manuscripts utilizing custom algorithms or software that are central to the research but not yet described in published literature, software must be made available to editors and reviewers. We strongly encourage code deposition in a community repository (e.g. GitHub). See the Nature Portfolio guidelines for submitting code & software for further information.

## Data

Policy information about availability of data

All manuscripts must include a data availability statement. This statement should provide the following information, where applicable:
- Accession codes, unique identifiers, or web links for publicly available datasets
- A description of any restrictions on data availability
- For clinical datasets or third party data, please ensure that the statement adheres to our policy

The data will be provided as a source data file with this article. Part of data has been deposited in public repositories (GitHub and Zenodo).

## Research involving human participants, their data, or biological material

Policy information about studies with human participants or human data. See also policy information about sex, gender (identity/presentation), and sexual orientation and race, ethnicity and racism.

| | |
|---|---|
| Reporting on sex and gender | *Use the terms sex (biological attribute) and gender (shaped by social and cultural circumstances) carefully in order to avoid confusing both terms. Indicate if findings apply to only one sex or gender; describe whether sex and gender were considered in study design; whether sex and/or gender was determined based on self-reporting or assigned and methods used.*<br>*Provide in the source data disaggregated sex and gender data, where this information has been collected, and if consent has been obtained for sharing of individual-level data; provide overall numbers in this Reporting Summary. Please state if this information has not been collected.*<br>*Report sex- and gender-based analyses where performed, justify reasons for lack of sex- and gender-based analysis.* |
| Reporting on race, ethnicity, or other socially relevant groupings | *Please specify the socially constructed or socially relevant categorization variable(s) used in your manuscript and explain why they were used. Please note that such variables should not be used as proxies for other socially constructed/relevant variables (for example, race or ethnicity should not be used as a proxy for socioeconomic status).*<br>*Provide clear definitions of the relevant terms used, how they were provided (by the participants/respondents, the researchers, or third parties), and the method(s) used to classify people into the different categories (e.g. self-report, census or administrative data, social media data, etc.)*<br>*Please provide details about how you controlled for confounding variables in your analyses.* |
| Population characteristics | *Describe the covariate-relevant population characteristics of the human research participants (e.g. age, genotypic information, past and current diagnosis and treatment categories). If you filled out the behavioural & social sciences study design questions and have nothing to add here, write "See above."* |
| Recruitment | *Describe how participants were recruited. Outline any potential self-selection bias or other biases that may be present and how these are likely to impact results.* |
| Ethics oversight | *Identify the organization(s) that approved the study protocol.* |

Note that full information on the approval of the study protocol must also be provided in the manuscript.

# Field-specific reporting

Please select the one below that is the best fit for your research. If you are not sure, read the appropriate sections before making your selection.

☒ Life sciences    ☐ Behavioural & social sciences    ☐ Ecological, evolutionary & environmental sciences

For a reference copy of the document with all sections, see nature.com/documents/nr-reporting-summary-flat.pdf

# Life sciences study design

All studies must disclose on these points even when the disclosure is negative.

| | |
|---|---|
| Sample size | Eta-squared is calculated to measure the effect size. The sample size of each experiment was determine according to previous publications (Refs:Science 329, 1075-1078; Front. Plant Sci. 8:1342; Elife 2, e00421. Nat. Comm. 14:3666). |
| Data exclusions | If there is data point differers significantly from other observations, the outlier was determined by using ROUT method build in GraphPad Prism 10 (Motulsky HJ and Brown RE., 2006, BMC Bioinformatics). |
| Replication | The replication numbers of each result are provided in figure legends. |
| Randomization | The plants and insects were randomly allocated into experimental groups. |
| Blinding | Not relevant to this study design since our experiments are quantitative measurements which did not require subjective interpretation or judgment. |

# Reporting for specific materials, systems and methods

We require information from authors about some types of materials, experimental systems and methods used in many studies. Here, indicate whether each material, system or method listed is relevant to your study. If you are not sure if a list item applies to your research, read the appropriate section before selecting a response.

## Materials & experimental systems

| n/a | Involved in the study |
|-----|----------------------|
| ☐ | ☒ Antibodies |
| ☒ | ☐ Eukaryotic cell lines |
| ☒ | ☐ Palaeontology and archaeology |
| ☐ | ☒ Animals and other organisms |
| ☒ | ☐ Clinical data |
| ☒ | ☐ Dual use research of concern |
| ☐ | ☒ Plants |

## Methods

| n/a | Involved in the study |
|-----|----------------------|
| ☒ | ☐ ChIP-seq |
| ☒ | ☐ Flow cytometry |
| ☒ | ☐ MRI-based neuroimaging |

## Antibodies

| | |
|---|---|
| Antibodies used | Anti-GST Antibody (B-14) HRP, Santa Cruz Biotechnology (sc-138 HRP) |
| Validation | GST Antibody (B-14) is a mouse monoclonal IgG1 κ GST antibody. The detail of validation has been provided on manufacturer website. https://www.scbt.com/p/gst-antibody-b-14 |

## Animals and other research organisms

Policy information about studies involving animals; ARRIVE guidelines recommended for reporting animal research, and Sex and Gender in Research

| | |
|---|---|
| Laboratory animals | Manduca sexta, Bombyx mori, Pieris rapae, Heliothis virescens, Danaus plexippus |
| Wild animals | Did not involve wild animals |
| Reporting on sex | n/a |
| Field-collected samples | The study did not involved in samples collected from filed. |
| Ethics oversight | No ethics approval or guidance was required. This research did not involved in any invertebrates or vulnerable groups. |

Note that full information on the approval of the study protocol must also be provided in the manuscript.

## Dual use research of concern

Policy information about dual use research of concern

### Hazards

Could the accidental, deliberate or reckless misuse of agents or technologies generated in the work, or the application of information presented in the manuscript, pose a threat to:

| No | Yes | |
|----|-----|---|
| ☒ | ☐ | Public health |
| ☒ | ☐ | National security |
| ☒ | ☐ | Crops and/or livestock |
| ☒ | ☐ | Ecosystems |
| ☒ | ☐ | Any other significant area |

## Experiments of concern

Does the work involve any of these experiments of concern:

| No | Yes | |
|----|-----|---|
| ☒ | ☐ | Demonstrate how to render a vaccine ineffective |
| ☒ | ☐ | Confer resistance to therapeutically useful antibiotics or antiviral agents |
| ☒ | ☐ | Enhance the virulence of a pathogen or render a nonpathogen virulent |
| ☒ | ☐ | Increase transmissibility of a pathogen |
| ☒ | ☐ | Alter the host range of a pathogen |
| ☒ | ☐ | Enable evasion of diagnostic/detection modalities |
| ☒ | ☐ | Enable the weaponization of a biological agent or toxin |
| ☒ | ☐ | Any other potentially harmful combination of experiments and agents |

# Plants

| | |
|---|---|
| Seed stocks | White mulberry seeds were purchased from Saflax.de. Tomato and milkweed plants are part of the stock in the greenhouse at the University of Amsterdam. |
| Novel plant genotypes | *Describe the methods by which all novel plant genotypes were produced. This includes those generated by transgenic approaches, gene editing, chemical/radiation-based mutagenesis and hybridization. For transgenic lines, describe the transformation method, the number of independent lines analyzed and the generation upon which experiments were performed. For gene-edited lines, describe the editor used, the endogenous sequence targeted for editing, the targeting guide RNA sequence (if applicable) and how the editor was applied.* |
| Authentication | *Describe any authentication procedures for each seed stock used or novel genotype generated. Describe any experiments used to assess the effect of a mutation and, where applicable, how potential secondary effects (e.g. second site T-DNA insertions, mosiacism, off-target gene editing) were examined.* |

