## [Peer Review File · Nature Ecology & Evolution]

Convergent evolution of (3Z):(2E)-hexenal isomerase in Lepidoptera and plants

Corresponding Author: Dr Yu-Hsien Lin

Version 0:

Decision Letter:

24th September 2025

Dear Dr Lin,

Your manuscript entitled "FAD-dependent hexenal isomerases in Lepidoptera evolved convergently with plant-derived hexenal isomerases" has now been seen by 3 reviewers, whose comments are attached. The reviewers have raised a number of concerns which will need to be addressed before we can offer publication in Nature Ecology & Evolution. We will therefore need to see your responses to the criticisms raised and to some editorial concerns, along with a revised manuscript, before we can reach a final decision regarding publication.

We therefore invite you to revise your manuscript taking into account all reviewer and editor comments. Please highlight all changes in the manuscript text file [OPTIONAL: in Microsoft Word format].

* If you have not done so already please begin to revise your manuscript so that it conforms to our Article format instructions at <http://www.nature.com/natecolevol/info/final-submission>. Refer also to any guidelines provided in this letter.

* Extended Data Figures - please ensure that any supplementary figures and tables that are crucial to the manuscript's conclusions are converted into Extended Data figures and tables to increase visibility of these data. Extended Data figures and tables are online-only (present in the online PDF and full-text HTML versions of the paper), peer-reviewed display items that provide essential background to the article but are not included in the main article due to space constraints. A maximum of ten Extended Data display items (figures and tables) is permitted.

Link Redacted

Nature Ecology & Evolution is committed to improving transparency in authorship. As part of our efforts in this direction, we are now requesting that all authors identified as 'corresponding author' on published papers create and link their Open Researcher and Contributor Identifier (ORCID) with their account on the Manuscript Tracking System (MTS), prior to acceptance. ORCID helps the scientific community achieve unambiguous attribution of all scholarly contributions. You can create and link your ORCID from the home page of the MTS by clicking on 'Modify my Springer Nature account'. For more information please visit www.springernature.com/orcid.

Yours sincerely,

[redacted]

Reviewer expertise:

Reviewer #1: plant-insect interactions, plant volatiles

Reviewer #2: plant metabolism, GC/MS

Reviewer #3: plant-insect coevolution, insect genomics

Reviewers' comments:

Reviewer #1 (Remarks to the Author):

This highly original study investigated the functional diversification and evolutionary origin of hexenal isomerases (Hi) in Lepidoptera, which convert Z-3-hexenal to E-2-hexenal. By changing the profiles of these green leaf volatiles (GLVs) emissions from plants Hi can affect their ecological function and exploitation. The three main findings are:

- Phylogenetic analysis revealed that Hi homologs in Lepidoptera belong to the GMC oxidoreductase family, whereas the Hi in plants are shown to involve a distinct GMC β subclade that is enriched in Hi homologs.
- The authors found species-specific Hi activity, with *Manduca sexta* Hi-1 exhibiting the highest efficiency in converting Z-3-hexenal to E-2-hexenal, both in vitro and in planta. Structural modeling identified key FAD-binding residues critical for Hi activity, and mutagenesis confirmed their essential role in the catalysis.
- In addition, comparative phylogenetics showed that Hi enzymes in plants and Lepidoptera evolved independently during the Cretaceous angiosperm radiation, implying a case of functional convergence. Taken together, the study highlights the ecological and physiological significance of Hi in modulating plant-insect interactions and suggests its role in herbivore adaptation to host plants.

The manuscript is nicely written and the results are convincing and conclusions valid. The introduction and discussion could be clearer about the proposed functions of the Hi genes.

What follows are a number of minor criticisms and suggestions for improvement of the text.

TITLE

I am not sure if using the acronym FAD is appropriate for the title.

INTRODUCTION

You may want to cite some of the original studies on the role of GLV's in direct defense and tritrophic interactions, as well as in plant defense priming and/or induction.

Line 54: What do you mean with "signaling capacity" and what is an example of E2AL's signaling effect?

Line 57: Here you create the impression that Hi has a highly specific function in the re-arrangement from Z3AL to E2AL. Later (in the discussion mainly) you point out that it is very likely also involved in similar conversions of other plant-produced compounds (concluded from the findings by Lin et al., 2023). It might be good to briefly point out that a physiological role rather than an ecological role could be or have been Hi's primary function.

Line 65: Strange that *Manduca* moths are attracted by a volatile that indicates a plant is already under attack. Could the effect be dose-dependent?

Fig 1 b-e: You may consider to use the same x-axis scale in each figure to better visualize the relatively higher expression in *M. sexta*.

RESULTS

Line 103: The use of "Notably," throughout the text is not always relevant (like here).

You may want to comment on the fact that E-2-hexenal shows up in all chromatograms (also in controls). This is because the standard is not pure or spontaneous partial conversion to Z-3-hexenal during the measurements. In the Materials and Methods you may want to comment on the spontaneous conversion of Z-3-hexenal that was used (which could exclude the impurity argument).

DISCUSSION

Lines 243-244: It might be less confusing to write: "...indicating that in various lepidopteran species these enzymes may contribute to the herbivore-driven production of E2AL."

Line 249: "overtime" should be "over time". You can also just delete it.

Lines 257-259: Here too it would be appropriate to cite some of the original studies on these effectors.

Line 324-337: Is the absence of a Hi gene in *P. xylostella* an indication that such genes are not really important for physiological processes, has suggested earlier?

Line 338: the generated chronogram...

MATERIALS AN METHODS

Line 157-160: You may want to cite the studies that identified the two fungal reference GMC oxidoreductases. I did not find a description of this procedure in the M&M.

REFERENCES

Double check for correct format and type scientific names in italics. Provide page numbers for the Jones et al. (2021) reference.

Reviewer #2 (Remarks to the Author):

In their article Lin et al study the convergent evolution of Lepidoptera and plant derived hexenal isomerases. I found the article to be a tour-de-force combining both traditional gene based analyses with contemporary alpha-fold analyses and functional enzyme activities. All are performed to excellent levels of technical quality and the paper is also excellently written.

One part of the paper which I do feel could be improved is the section where they state it is not clear whether HI (sorry but as an abbreviation both should be capitalized!) ultimately benefits the plant or the insect remains unclear as this depends on context. Could the authors expand on this fascinating observation - potentially including a model depicting this would be helpful. Other than this I really cannot fault the paper.

Reviewer #3 (Remarks to the Author):

This comprehensive study of hexenal isomerase (Hi) evolution in Lepidoptera shows that these enzymes evolved independently from plant Hi through convergent evolution, which is a fascinating cross-Kingdom story. To do this, they authors combine phylogenetic analysis across 34 lepidopteran species along with functional biochemistry and were able to identify FAD-dependent GMC oxidoreductases that convert Z-3-hexenal to E-2-hexenal. They found that Hi is phylogenetically restricted to Apoditrysia within the Lepidoptera and that there was species-specific variation in enzyme activity. Then then confirmed with site directed mutagenesis the critical residues for potential FAD binding and catalysis. The most salient findings apply only to *Manduca sexta*.

Overall, I found that this manuscript was well written, conceived of, and in providing valuable insights into plant-insect interactions that we can think of as "coevolution" in the broad sense, and separately on the biochemical basis of volatile compound manipulation, which is of interest to agricultural and ecological researchers. The convergence story is interesting and there have been a number of similar stories on plants and insects that produce the same specialized metabolites that suggest this is more common than we have appreciated as a field.

I did have constructive comments, both major and minor that I detail below.

Major Comments

1. The ecological significance needs to be narrowed:

The authors claim that Hi activity influences "multitrophic interactions", but this rests almost entirely on studies involving *Manduca sexta*. While *M. sexta* Hi-1 produces increases in E-2-hexenal emissions, the other three lepidopteran species tested (*C. virescens*, *D. plexippus*, *B. mori*) showed minimal activity in planta despite having functional enzymes in vitro. This suggests *M. sexta* may be exceptional rather than representative and this calls into question the broad ecological significance of the findings. The ecological claims should be tempered and, I think, limited to the *M. sexta* system, which is fine.

2. The mechanistic understanding of FAD requirement is incomplete and this needs to be acknowledged:

The authors propose that FAD enables a non-redox isomerization reaction, which is biochemically unusual. While they demonstrate that FAD enhances activity and that mutations to predicted FAD-binding residues reduce function, they don't show direct FAD binding, identify intermediates, or adequately explain why this cofactor is required for an acid-base catalyzed isomerization. The proposed mechanism comparing to polyunsaturated fatty acid isomerase seems speculative. I think this should be reframed as an empirical observation requiring further study rather than a resolved mechanism.

3. There are some experimental design inconsistencies that require explanation:

It seems that the protein concentrations they used vary among species without an explanation of why in the main text. Although on the one hand this makes sense, the testing of each species on its preferred host plant ends up confounding the species-specific enzyme activities with any plant-species specific factors that might influence Hi function. I'm not sure what the right way to deal with this is, but I encourage the authors to address this.

4. Figure accessibility and statistical reporting details need some revision:

Many of the figures use red-green color schemes that would be difficult for red-green colorblind readers to interpret. For example, Figure 1a's orange highlighting, Figure 3's red-green pie charts, and Figure 5a's red clade designation should use colorblind-friendly palettes or additional visual cues, and the authors should take a careful look at the rest of their figures. I also noticed that many of the comparisons rely on relatively small sample sizes which is fine in many cases (standard in molecular biology to have 3 biological replicates) but they only report p-values without any effect sizes or at times confidence intervals/error bars are not explained in the figure caption, making it difficult to assess biological significance beyond statistical significance.

Minor Comments

Line 22: "reshapes GLV profiles and influences multi-trophic interactions" should be qualified as "may influence multi-trophic interactions" since this is only demonstrated in *M. sexta*.

Line 28: "*Manduca sexta* Hi-1 displaying the highest activity under identical protein concentrations" is confusing because the methods state that different protein concentrations were used for different species (although I might be confused by this). If I am not confused then this should read "displaying the highest specific activity among tested homologs."

Line 32: "during the Cretaceous angiosperm radiation" implies causation. Change to "coinciding with the Cretaceous angiosperm radiation."

Line 50: "becomes far more efficient" needs quantification with actual fold-change data.

Line 61: "secrete a functionally analogous" should be "produce in their oral secretions".

Line 127: "MsHi-1 in *M. sexta* appears uniquely efficient" is a bit problematic given that what the optimal conditions are may differ between species.

Line 144: "does not involve a net redox change" should clarify why the FAD requirement is biochemically unexpected for readers unfamiliar with this.

Line 164: "correspond to a conserved histidine (H135)" needs quantification of how many sequences show this conservation.

Line 189: "Both plant- and Lepidopteran-derived Hi display promiscuous activity toward other Z-3-aldehydes" - this broad substrate specificity deserves more discussion regarding evolutionary implications and the L should be lowercase since it is an adjective.

Line 252: That specialist herbivores *D. plexippus* and *B. mori* show much lower activity seems to contradict the broad ecological importance claims and deserves more discussion.

Line 276: "may resemble its function" is speculative and the authors should note that direct evidence is lacking.

Line 291: The proposed mechanism throughout this section needs more qualifying language since it's largely speculative.

Line 314: "higher proportion of E2AL compared to Z3AL" should provide actual ratios rather than qualitative descriptions.

Line 508-510: The differences in protein concentrations used (0.0625 µg for *M. sexta* vs. 3 µg for *B. mori*) need justification in the main text.

Line 542: "Different letters above bars in the graphs indicate statistically significant differences" - this should be explained in each figure caption.

Line 575-594 (Figure 1 caption): The authors should explain why different protein concentrations were used for different species in panels b-e.

Line 599 (Figure 2a caption): "identical protein concentrations" just double check all of the concentration statements.

Line 621: The reports of F-statistics are good but we also need the magnitude of differences between treatments - effect sizes are also needed.

*****END*****

Version 1:

Decision Letter:

10th November 2025

Dear Yu-Hsien,

Thank you for submitting your revised manuscript "Flavin-dependent hexenal isomerases in Lepidoptera evolved convergently with plant-derived hexenal isomerases" (NATECOLEVOL-25072396A). It has now been seen again by the original reviewers and their comments are below. The reviewers find that the paper has improved in revision, and therefore we'll be happy in principle to publish it in *Nature Ecology & Evolution*, pending minor revisions to satisfy the reviewers' final requests and to comply with our editorial and formatting guidelines.

If you have not done so already, please ensure that you also email us a completed copy of the Reporting summary :

Reporting summary: https://www.nature.com/documents/nr-reporting-summary.pdf

Thank you again for your interest in *Nature Ecology & Evolution*. Please do not hesitate to contact me if you have any questions.

Sincerely,

[redacted]

Reviewer #1 (Remarks to the Author):

The manuscript has been very nicely revised in accordance with the reviewers' comments and recommendations. I have no further concerns or suggestions for improvement. The only suggestion:

- Consistently use "lepidopteran" without capital "L" throughout the text. (Lepidoptera should be with capital).
- I would write "in planta" in italics (as it is Latin) throughout the text.

Reviewer #2 (Remarks to the Author):

I am 100 percent satisfied to the reviewers responses to not only my original review but also to the other reviewers comments. The authors should be commended on the thorough job they did here.

Reviewer #3 (Remarks to the Author):

The authors did a really great job responding to my comments (and catching an error I made regarding whether a plot had protein vs. RNA) and to those of the other reviewers. I only ask that they change Fig. 4f through g' so that there isn't red and green together otherwise they will lose colorblind readers. This is a terrific contribution to the literature.

Reviewers' comments:

Reviewer #1 (Remarks to the Author):

This highly original study investigated the functional diversification and evolutionary origin of hexenal isomerases (Hi) in Lepidoptera, which convert Z-3-hexenal to E-2-hexenal. By changing the profiles of these green leaf volatiles (GLVs) emissions from plants Hi can affect their ecological function and exploitation. The three main findings are:

- Phylogenetic analysis revealed that Hi homologs in Lepidoptera belong to the GMC oxidoreductase family, whereas the Hi in plants are shown to involve a distinct GMC β subclade that is enriched in Hi homologs.
- The authors found species-specific Hi activity, with *Manduca sexta* Hi-1 exhibiting the highest efficiency in converting Z-3-hexenal to E-2-hexenal, both in vitro and in planta. Structural modeling identified key FAD-binding residues critical for Hi activity, and mutagenesis confirmed their essential role in the catalysis.
- In addition, comparative phylogenetics showed that Hi enzymes in plants and Lepidoptera evolved independently during the Cretaceous angiosperm radiation, implying a case of functional convergence. Taken together, the study highlights the ecological and physiological significance of Hi in modulating plant-insect interactions and suggests its role in herbivore adaptation to host plants.

The manuscript is nicely written and the results are convincing and conclusions valid. The introduction and discussion could be clearer about the proposed functions of the Hi genes.

What follows are a number of minor criticisms and suggestions for improvement of the text.

Ans: We appreciate the reviewer's positive feedback and recognition of our work. We have carefully revised the manuscript following the reviewer#1's comments. All corresponding modifications have been highlighted in yellow in the revised manuscript.

TITLE

I am not sure if using the acronym FAD is appropriate for the title.

Ans: We agree and have revised the title to avoid the acronym "FAD." The new title reads: Flavin-dependent hexenal isomerases in Lepidoptera evolved convergently with plant-derived hexenal isomerases

INTRODUCTION

You may want to cite some of the original studies on the role of GLV's in direct defense and tritrophic interactions, as well as in plant defense priming and/or induction.

Ans: We have added foundational references documenting the roles of GLVs in direct and indirect defense, tritrophic interactions, and defense priming in lines 41.

Line 54: What do you mean with "signaling capacity" and what is an example of E2AL's signaling effect?

Ans: We have rephrased the sentences and added an example in line 51-56.

“Recent studies showed that E2AL triggers a calcium burst and activates the WRKY46–MYC2 transcriptional module in Arabidopsis, thereby promoting flavonoid accumulation and enhancing anti-herbivore defenses (Hao et al., 2024).”

Line 57: Here you create the impression that Hi has a highly specific function in the re-arrangement from Z3AL to E2AL. Later (in the discussion mainly) you point out that it is very likely also involved in similar conversions of other plant-produced compounds (concluded from the findings by Lin et al., 2023). It might be good to briefly point out that a physiological role rather than an ecological role could be or have been Hi’s primary function.

Ans: We have added a sentence to point this out (line 65-68).

“While Z3AL to E2AL conversion is the well-characterized reaction, both lepidopteran and plant Hi can also act on related Z-3-alkenals such as Z-3-octenal or Z-3-nonenal, and may fulfill physiological roles in lepidoptera in addition to their ecological functions (Kunishima et al., 2016; Lin et al., 2023).”

Line 65: Strange that Manduca moths are attracted by a volatile that indicates a plant is already under attack. Could the effect be dose-dependent?

Ans: We thank the reviewer for the comment. Our original sentence may have caused some confusion. Based on field studies, Manduca females are not attracted to plants emitting high levels of E2AL per se. Instead, they preferentially oviposit on plants with GLV blends enriched in Z-3-GLVs (high Z/E ratio), thereby avoiding hosts with high E-2-GLVs that indicate the presence of conspecific larvae and potential competition (Allmann et al., 2013). Furthermore, the ecological relevance of E2AL in attracting natural enemies such as Geocoris predators also depends on the shift in the Z/E ratio, rather than the absolute dose of E2AL (Allmann et al., 2010). We have revised the text as follows to clarify this point (line 64)

“Such conversion alters the ratio of Z-3/E-2-GLVs, which guides female moths in choosing oviposition sites (Allmann et al., 2013), but paradoxically also serves as the cue that attracts their natural enemies (Allmann and Baldwin, 2010).”

Fig 1 b-e: You may consider to use the same x-axis scale in each figure to better visualize the relatively higher expression in *M. sexta*.

Ans: We thank the reviewer for this suggestion. In Fig. 1b–e, the expression values were normalized within each species relative to the tissue with the lowest expression value of that species. This normalization allowed us to visualize tissue-specific differences per species but means that absolute values are not directly comparable across species. We have revised the Fig. 1 legend to clarify this point (line 592).

RESULTS

Line 103: The use of “Notably,” throughout the text is not always relevant (like here).

Ans: We have removed or revised nonessential uses of “Notably” as follows to improve readability:

~~Notably~~, in vitro assays reveal that Hi activity varies among Lepidopteran species.” (line 70)

~~Notably~~, oral secretions (OS) of these four species had previously been shown to exhibit Hi activity.” (line 103)

~~Notably~~ Moreover, suppression of GLVs via FHD has been shown to decrease parasitoid attraction, potentially..... (line 256)

~~Notably~~ Consistent with this pattern, several adaptive traits in Apoditrysia, such as a versatile proboscis and expanded detoxification gene families.....(line 349)

~~Notably~~, all analyzed species from both Brassicaceae and Caricaceae families....(line 683)

You may want to comment on the fact that E-2-hexenal shows up in all chromatograms (also in controls). This is because the standard is not pure or spontaneous partial conversion to Z-3-hexenal during the measurements. In the Materials and Methods you may want to comment on the spontaneous conversion of Z-3-hexenal that was used (which could exclude the impurity argument).

Ans: We thank the reviewer for pointing this out. As suggested, we have clarified this in the Materials and Methods as follows (line 500-504):

“Low background levels of *E*-2-hexenal were consistently detected in each sample as a result of the spontaneous non-enzymatic conversion of *Z*-3-hexenal during measurements. To correct for this, the calculated value was adjusted by subtracting the non-enzymatic conversion rate determined from a buffer-only control.....”

DISCUSSION

Lines 243-244: It might be less confusing to write: “...indicating that in various lepidopteran species these enzymes may contribute to the herbivore-driven production of E2AL.”

Ans: We have removed that sentence (line 241) in response to Reviewer #3’s major comment 1.

Line 249: “overtime” should be “over time”. You can also just delete it.

Ans: We have removed it. Now it reads:

...significantly higher levels of E2AL ~~overtime~~ compared to plants fed on by *C. virescens* (line 245-246)

Lines 257-259: Here too it would be appropriate to cite some of the original studies on these effectors.

Ans: We have expanded citations to original studies on these effectors (line 255).

Line 324-337: Is the absence of a Hi gene in *P. xylostella* an indication that such genes are not really important for physiological processes, has suggested earlier?

ANS: We thank the reviewer for raising this thoughtful point. We have expanded the discussion to note that *P. xylostella* lacks Hi without apparent fitness loss, whereas *Manduca* Hi-1 mutants show developmental defects. We further highlight that

Lepidoptera have undergone lineage-specific expansions of the GMC β subfamily (Iida et al., 2007; Sun et al., 2012), and such redundancy may reduce the selective constraint to retain Hi specifically, especially if related physiological functions are maintained by other GMC β paralogs. This clarification has been added at lines 331–339 in the revised manuscript.

Line 338: the generated chronogram...

Ans: We have added “the generated” as suggested (line 340).

MATERIALS AND METHODS

Line 157-160: You may want to cite the studies that identified the two fungal reference GMC oxidoreductases. I did not find a description of this procedure in the M&M.

Ans: We have included the citations in line 159-160

REFERENCES

Double check for correct format and type scientific names in italics. Provide page numbers for the Jones et al. (2021) reference.

Ans: We have corrected references and provided page numbers for Jones et al. (2021).

Reviewer #2 (Remarks to the Author):

In their article Lin et al study the convergent evolution of Lepidoptera and plant derived hexanal isomerases. I found the article to be a tour-de-force combining both traditional gene based analyses with contemporary alpha-fold analyses and functional enzyme activities. All are performed to excellent levels of technical quality and the paper is also excellently written.

One part of the paper which I do feel could be improved is the section where they state it is not clear whether HI (sorry but as an abbreviation both should be capitalized!) ultimately benefits the plant or the insect remains unclear as this depends on context. Could the authors expand on this fascinating observation - potentially including a model depicting this would be helpful. Other than this I really cannot fault the paper.

Ans: We thank the reviewer for the very positive evaluation of our study and the suggestions. We have revised the manuscript to adopt a dual nomenclature system that follows both plant and insect naming conventions: **HI** is used when referring exclusively to plant proteins, and **Hi** is used when referring to insect proteins, when describing both taxa together, or when mentioning enzyme activity (e.g., Hi activity). This nomenclature convention has been described in the methods section (line 362-368). We have made corresponding changes throughout the manuscript and supplementary information, remade Fig. 5, S6, and S7, and all modifications have been highlighted in **green** in the revised version.

Whether Hi ultimately benefits the plant or the insect is indeed an intriguing question, but currently difficult to answer conclusively and is beyond the main scope of this study, which focuses on the evolutionary origin and diversification of Hi proteins. From the perspective of GLV modulation, several reviews have discussed how shifts in GLV composition can have both positive and negative effects on either the plant or the

herbivore, depending on ecological context (Ameye et al., 2018; Matsui and Engelberth, 2022). Our previous work demonstrated that Hi in insects can also serve physiological functions unrelated to Z3AL isomerization, as Hi mutant *M. sexta* reared on volatile-free artificial diet show developmental defects. In plants, HI belongs to the cupin superfamily, members of which often serve diverse physiological roles such as seed storage or stress-related functions (Lakhssassi et al., 2024; Shutov et al., 2003; Smirnov and Arnaud, 2019). Consistent with this, our previous and present analyses revealed that several plant Hi homologs clustered within the Hi clade (Fig. 5a) are lack of Hi catalytic residues and activity, suggesting possible functional divergence within the clade. However, because no *HI* mutant plants have yet been characterized, the physiological roles of plant HI remain unresolved. Therefore, while both plant and insect Hi likely perform additional functions beyond Z3AL isomerization, current data are insufficient to determine whether Hi activity ultimately benefits the plant or the insect.

Reviewer #3 (Remarks to the Author):

This comprehensive study of hexenal isomerase (Hi) evolution in Lepidoptera shows that these enzymes evolved independently from plant Hi through convergent evolution, which is a fascinating cross-Kingdom story. To do this, they authors combine phylogenetic analysis across 34 lepidopteran species along with functional biochemistry and were able to identify FAD-dependent GMC oxidoreductases that convert Z-3-hexenal to E-2-hexenal. They found that Hi is phylogenetically restricted to Apoditrysia within the Lepidoptera and that there was species-specific variation in enzyme activity. Then then confirmed with site directed mutagenesis the critical residues for potential FAD binding and catalysis. The most salient findings apply only to *Manduca sexta*.

Overall, I found that this manuscript was well written, conceived of, and in providing valuable insights into plant-insect interactions that we can think of as "coevolution" in the broad sense, and separately on the biochemical basis of volatile compound manipulation, which is of interest to agricultural and ecological researchers. The convergence story is interesting and there have been a number of similar stories on plants and insects that produce the same specialized metabolites that suggest this is more common than we have appreciated as a field.

I did have constructive comments, both major and minor that I detail below.

Ans: We sincerely thank reviewer #3 for the positive assessment of our work. We have carefully considered all points raised and revised the manuscript accordingly. The changes made in response to the reviewer's comments are highlighted in blue in the revised manuscript. A detailed point-by-point response is provided below.

Major Comments

1. The ecological significance needs to be narrowed:

The authors claim that Hi activity influences "multitrophic interactions", but this rests almost entirely on studies involving *Manduca sexta*. While *M. sexta* Hi-1 produces increases in E-2-hexenal emissions, the other three lepidopteran species tested (*C. virescens*, *D. plexippus*, *B. mori*) showed minimal activity in planta despite having functional enzymes in vitro. This suggests *M. sexta* may be exceptional rather than representative and this calls into question the broad ecological significance of the findings. The ecological claims should be tempered and, I think, limited to the *M. sexta* system, which is fine.

Ans: We agree and have tempered our ecological claims as follows to avoid overgeneralization.

Abstract

A key mechanism underlying this plasticity is the conversion of Z-3-hexenal to E-2-hexenal by the enzyme (3Z):(2E)-hexenal isomerase (Hi), which reshapes GLV profiles and **may** influences multi-trophic interactions. (line 23)

Discussion:

Functional assays of Lepidopteran Hi homologs revealed that at least one gene of each examined species exhibited measurable Hi activity in vitro, ~~indicating that in various lepidopteran species these enzymes may contribute to the herbivore-driven production of E2AL.~~ (line 240-241)

The ecological relevance of Hi is well established **in the *M. sexta* system**,...(Line 250)

2. The mechanistic understanding of FAD requirement is incomplete and this needs to be acknowledged:

The authors propose that FAD enables a non-redox isomerization reaction, which is biochemically unusual. While they demonstrate that FAD enhances activity and that mutations to predicted FAD-binding residues reduce function, they don't show direct FAD binding, identify intermediates, or adequately explain why this cofactor is required for an acid-base catalyzed isomerization. The proposed mechanism comparing to polyunsaturated fatty acid isomerase seems speculative. I think this should be reframed as an empirical observation requiring further study rather than a resolved mechanism.

Ans: We thank the reviewer for this comment. As suggested, we have clarified in the Results that the conversion of Z3AL to E2AL represents a cis–trans rearrangement, and we now note in the discussion that direct experimental evidence for this mechanism in Hi is currently lacking.

The revised sentence in Result (line 144-147) now reads:

“Although the conversion of Z3AL to E2AL represents a cis–trans rearrangement without a net redox change, we hypothesized that Lepidopteran Hi still requires FAD for its isomerase activity, because the FAD-binding site constitutes a conserved catalytic feature of GMC oxidoreductases.”

The revised sentence in Discussion (line 273-274) now reads:

“While the precise role of FAD in Hi catalysis remains unresolved, one possibility is that it functions analogously to polyunsaturated fatty acid isomerase (PAI) from”

3. There are some experimental design inconsistencies that require explanation:

It seems that the protein concentrations they used vary among species without an explanation of why in the main text. Although on the one hand this makes sense, the testing of each species on its preferred host plant ends up confounding the species-specific enzyme activities with any plant-species specific factors that might influence Hi function. I'm not sure what the right way do deal with this is, but I encourage the authors to address this.

Ans: We thank the reviewer for this comment. We would like to clarify that all homologs were assayed at the same set of three protein concentrations (0.01, 0.1, and 1 µg) in the

activity assays shown in Fig. 2, and cross-species comparisons were always made at identical protein quantities. Only in the FAD supplementation assay (Fig. 4) were different protein amounts used. The protein concentrations in the FAD supplementation assay were deliberately adjusted for each species in order to prevent complete substrate (*Z*-3-hexenal) conversion in highly active enzymes (e.g. MsHi-1) and to ensure detectable activity in weakly active ones. This assay was designed to evaluate the effect of FAD addition within each enzyme, not to compare activity levels across species. We described the rationale in the Methods (line 510–512).

However, to further improve clarity in the main text, we have revised the sentence to read: “MsHi-1 displayed the highest activity among tested homologs under identical protein concentrations” (line 28-29).

4. Figure accessibility and statistical reporting details need some revision:

Many of the figures use red-green color schemes that would be difficult for red-green colorblind readers to interpret. For example, Figure 1a's orange highlighting, Figure 3's red-green pie charts, and Figure 5a's red clade designation should use colorblind-friendly palettes or additional visual cues, and the authors should take a careful look at the rest of their figures. I also noticed that many of the comparisons rely on relatively small sample sizes which is fine in many cases (standard in molecular biology to have 3 biological replicates) but they only report p-values without any effect sizes or at times confidence intervals/error bars are not explained in the figure caption, making it difficult to assess biological significance beyond statistical significance.

Ans: We thank the reviewer for pointing this out.

Color accessibility:

We appreciate the reviewer's concern regarding figure accessibility. We suspect that some of the perceived red-green contrasts may stem from differences in screen calibration or print color reproduction, as the figures do not in fact use a red-green scheme. For instance, Figure 3 contains green-blue pie charts rather than red-green. Nevertheless, we carefully evaluated all figures for accessibility using the Colorblind Image Tester (https://bioapps.byu.edu/colorblind_image_tester) to assess readability for moderate-to-severe red-green colorblindness (deuteranopia). The results showed high confidence values, indicating that the figures are colorblind-friendly (Fig. 1: 100%; Fig. 2: 99.95%; Fig. 3: 100%; Fig. 4: 99.89%; Fig. 5: 96.56%). In addition, we ensured that figure legends are not only color-coded but also supported by textual labels. For example, in Fig. 5a the Hi clade is both highlighted in color and annotated in the figure. We are, of course, happy to further adjust figure colors if this would improve accessibility or clarity.

Error bars and effect size presentation:

In the original manuscript, error bars were described only in the Methods and in the Fig. 1 legend. To improve clarity and consistency, we have now stated “Error bars are presented as mean values \pm SD” in the legends of Fig. 4, Supplementary Fig. 2, and Supplementary Fig. 5. In addition, we emphasized in the Fig. 3 legend (pie charts) that Supplementary Fig. 2 shows the corresponding dot plots of all biological replicates, which illustrate the distribution and magnitude of treatment effects. We have also added the effect sizes (Eta-squared values) to the results of parametric tests, in the legend of Fig. 3, Fig. 4, Fig S2 and Fig S5.

Minor Comments

Line 22: "reshapes GLV profiles and influences multi-trophic interactions" should be qualified as "may influence multi-trophic interactions" since this is only demonstrated in *M. sexta*.

Ans: This sentence has been removed in response to the reviewer's first major comment.

Line 28: "Manduca sexta Hi-1 displaying the highest activity under identical protein concentrations" is confusing because the methods state that different protein concentrations were used for different species (although I might be confused by this). If I am not confused then this should read "displaying the highest specific activity among tested homologs."

Ans: A similar concern has been addressed in Major Comment 3.

Line 32: "during the Cretaceous angiosperm radiation" implies causation. Change to "coinciding with the Cretaceous angiosperm radiation."

Ans: We have revised the sentence to "coinciding with the Cretaceous angiosperm radiation," as suggested (line 34).

Line 50: "becomes far more efficient" needs quantification with actual fold-change data.

Ans: We have revised the wording to avoid overstatement. The phrase "far more efficient" has been tempered to "more efficient" (line 57). We think that it would not be appropriate to provide a quantitative fold-change estimate here, as the extent of non-enzymatic conversion strongly depends on external abiotic factors (e.g., temperature, pH, solvent conditions). Consequently, the apparent difference in conversion rates can vary substantially, and presenting a single numerical value could be misleading.

Line 61: "secrete a functionally analogous" should be "produce in their oral secretions".

Ans: We have revised the sentence as suggested. It now reads "...larvae of the hawk moth (*Manduca sexta*) produce a functionally analogous but phylogenetically distinct Hi protein in their oral secretions...." (line 61-62).

Line 127: "MsHi-1 in *M. sexta* appears uniquely efficient" is a bit problematic given that what the optimal conditions are may differ between species.

Ans: We thank the reviewer for pointing this out. We agree that the original phrasing may have overstated the conclusion. We have now revised the sentence to clarify that our inference is based on the tested experimental conditions. The revised sentence reads: "... MsHi-1 in *M. sexta* shows comparatively high efficiency in converting Z3AL into E2AL under the tested conditions, which in turn led to a pronounced increase in E-2-GLVs in our in planta assays" (line 139-141).

Line 144: "does not involve a net redox change" should clarify why the FAD requirement is biochemically unexpected for readers unfamiliar with this.

Ans: Addressed as part of Major Comment 2. We have clarified in the Results that the conversion of Z3AL to E2AL is a cis–trans rearrangement that does not involve a net redox change, which makes the FAD requirement unexpected. (line 144-147)

Line 164: "correspond to a conserved histidine (H135)" needs quantification of how many sequences show this conservation.

Ans: We thank the reviewer for this suggestion. All five Lepidopteran Hi homologs with measurable enzymatic activity in Fig. 2 contain this conserved histidine (Supplementary Fig. 3). We have revised the text to specify the number of homologs that retain this conserved residue. The sentence now reads:

".....were found to correspond to a conserved histidine (H135) in MsHi-1 (Fig. 4f, g') and in all four other Hi homologs that exhibit Hi activity (Fig. 2 and Supplementary Fig. 3)." (lines 164–165)

Line 189: "Both plant- and Lepidopteran-derived Hi display promiscuous activity toward other Z-3-aldehydes" - this broad substrate specificity deserves more discussion regarding evolutionary implications and the L should be lowercase since it is an adjective.

Ans: The letter L in *lepidopteran*-derived Hi has been changed to lowercase (line 188).

A previous study with partially purified cucumber Hi showed conversion across several Z-3-unsaturated aldehydes (C6–C9) and even the rearrangement of the fatty acid–derived 12-oxo-cis-9-dodecenoic acid to 12-oxo-trans-10-dodecenoic acid under *in vitro* conditions (Phillips et al., 1979). This pattern indicates that Hi recognizes the double bond at the Z-3- position rather than a specific chain length or the presence of a terminal aldehyde group. Accordingly, activity toward diverse Z-3-aldehydes likely reflects an inherent property of the catalytic mechanism rather than an adaptation to multiple physiological substrates.

Our intention in this statement was to describe the *in vitro* substrate range of Hi rather than to imply multiple physiological primary substrates. Enzyme promiscuity is common across protein families and provides starting points for the evolution of new activities (Copley, 2015; Tawfik and S., 2010). However, whether a compound represents a biologically relevant substrate depends on catalytic efficiency (e.g., K_m , k_{cat}) and *in vivo* concentration. In previous studies, kinetic parameters of both plant and insect Hi were determined mainly for Z-3-hexenal, whereas reactions with longer-chain Z-3-alkenals were not kinetically quantified (Kunishima et al., 2016; Lin et al., 2023). Hence, the observed conversions of other Z-3-aldehydes may simply reflect general catalytic flexibility rather than broad physiological specificity.

To prevent any misinterpretation, we have adopted more conservative wording in line 189:

"Both plant- and lepidopteran-derived Hi display promiscuous activity toward other Z-3-aldehydes **in vitro**."

Line 252: That specialist herbivores *D. plexippus* and *B. mori* show much lower activity seems to contradict the broad ecological importance claims and deserves more discussion.

Ans: We thank the reviewer for pointing this out. As addressed in Major Comment 1, we have tempered our ecological claims to avoid overgeneralization beyond the *M. sexta* system.

Upon re-evaluating our manuscript, we acknowledge that our previous statement “Hi activity is not universally linked to trophic specialization” is not strongly supported by the current evidence, given that *D. plexippus* and *B. mori* are monophagous while *M. sexta* is oligophagous, representing different types of specialists. We have therefore removed this sentence and rephrased the corresponding part of the Discussion (lines 246–249).

Line 276: "may resemble its function" is speculative and the authors should note that direct evidence is lacking.

Ans: Addressed as part of Major Comment 2. We have revised the text as follows to note that direct experimental evidence for this mechanism in Hi is currently lacking. (line 292-293)

Line 291: The proposed mechanism throughout this section needs more qualifying language since it's largely speculative.

Ans: We have revised the paragraph in the Discussion to temper the language and emphasize that the proposed mechanism remains speculative. The revised sentences are as follows:

line 284-288

The Z3AL isomerization by plant HI has previously been ~~reported~~ **proposed** to involve a keto-enol tautomerism mechanism (Kunishima et al., 2016). The process is mediated by a conserved catalytic histidine (Fig. 5b), which **could** abstract a proton from the C2 position of Z3AL, forming a transient enolate intermediate. This enolate structure **may** allow the electron density to shift, facilitating the formation of a keto-like tautomer.

Line 289-293:

As this conserved histidine has been previously reported in other GMC oxidoreductases to abstract the proton from the substrate (Liu et al., 2013; Yoshida et al., 2015), this histidine may perform a role that is similar to a catalytic histidine in plant HI. **However, direct biochemical evidence confirming its involvement in the Z3AL isomerization remains to be established.**

Line 314: "higher proportion of E2AL compared to Z3AL" should provide actual ratios rather than qualitative descriptions.

Ans: We have revised the text in the Discussion to include quantitative information from Engelberth and Engelberth (2020). The sentence now reads:

“A previous study has shown that mechanical damage in Cucurbitaceae and Fabaceae species leads to the emission of E2AL as the dominant GLV aldehyde, with relative proportions of ~63% to 87% of the total aldehyde pool (Engelberth and Engelberth, 2020).” (line 309-311)

Line 508-510: The differences in protein concentrations used (0.0625 µg for *M. sexta* vs. 3 µg for *B. mori*) need justification in the main text.

Ans: A similar concern has been addressed in Major Comment 3.

To improve clarity, we have added the explanation in the caption of Fig. 4 as follows: "The amounts of protein used for each enzyme (0.0625 µg MsHi-1; 1 µg CvHi-1; 1.38 µg DpHi-1; 3 µg BmHi-1) were adjusted to avoid complete conversion of Z-3-hexenal, thereby enabling the effect of FAD addition to be evaluated" (Line 642-644).

Line 542: "Different letters above bars in the graphs indicate statistically significant differences" - this should be explained in each figure caption.

Ans: We thank the reviewer for this comment. We would like to clarify that only Fig. 3 and Fig. 4i included ANOVA followed by post hoc tests, and therefore these are the only figures with different letters indicating statistically significant differences. We have confirmed that the explanation is indeed provided in the captions of Fig. 3 (line 629-630) and Fig. 4i (line 665).

Line 575-594 (Figure 1 caption): The authors should explain why different protein concentrations were used for different species in panels b-e.

Ans: We thank the reviewer for the comment. We would like to clarify that panels b-e of Fig. 1 shows relative gene expression levels across different tissues (fat body, midgut, and salivary glands) obtained by qPCR. No recombinant proteins were used in these panels.

Line 599 (Figure 2a caption): "identical protein concentrations" just double check all of the concentration statements.

Ans: We thank the reviewer for this careful check. We would like to clarify that in Fig. 2a, all recombinant proteins were tested at the same set of three concentrations (0.01, 0.1, and 1 µg), and cross-species comparisons were always made at identical protein quantities. Only in the FAD supplementation assay (Fig. 4 a-e) we did use different protein amounts, and this was done solely to avoid complete substrate conversion and to allow us to assess the effect of FAD addition. These assays were not used for cross-species comparisons. To prevent confusion, we have added a sentence in the figure 2 captions. "Each protein was tested at the same three concentrations to assess concentration-dependent activity." (lines 606-607).

Line 621: The reports of F-statistics are good but we also need the magnitude of differences between treatments - effect sizes are also needed.

Ans: We thank the reviewer for this suggestion. We would like to note that the magnitude of differences between treatments has been visualized in the dot plots corresponding to Fig. 3, which display all biological replicates and their distribution. These plots are provided in Supplementary Fig. 2, as also indicated in the Fig. 3 legend. In addition, we have now added the effect sizes (Eta-squared values) to the results of the parametric tests, in the figure legend of Fig. 3, Fig. 4, Fig. S2 and Fig. S5.

References

- Allmann, S., Halitschke, R., Schuurink, R.C., Baldwin, I.T., 2010. Oxylin channelling in *Nicotiana attenuata*: lipoxygenase 2 supplies substrates for green leaf volatile production. *Plant, Cell & Environment* 33, 2028-2040.
- Allmann, S., Spathe, A., Bisch-Knaden, S., Kallenbach, M., Reinecke, A., Sachse, S., Baldwin, I.T., Hansson, B.S., 2013. Feeding-induced rearrangement of green leaf volatiles reduces moth oviposition. *Elife* 2, e00421.
- Ameys, M., Allmann, S., Verwaeren, J., Smagghe, G., Haesaert, G., Schuurink, R.C., Audenaert, K., 2018. Green leaf volatile production by plants: a meta-analysis. *New Phytologist* 220, 666-683.
- Copley, S.D., 2015. An evolutionary biochemist's perspective on promiscuity. *Trends in Biochemical Sciences* 40, 72-78.
- Kunishima, M., Yamauchi, Y., Mizutani, M., Kuse, M., Takikawa, H., Sugimoto, Y., 2016. Identification of (Z)-3:(E)-2-Hexenal Isomerases Essential to the Production of the Leaf Aldehyde in Plants. *Journal of Biological Chemistry* 291, 14023-14033.
- Lakhssassi, N., El Baze, A., Knizia, D., Salhi, Y., Embaby, M.G., Anil, E., Mallory, C., Lakhssassi, A., Meksem, J., Shi, H., Vuong, T.D., Meksem, K., Kassem, M.A., AbuGhazaleh, A., Nguyen, H.T., Bellaloui, N., Boualem, A., Meksem, K., 2024. A sucrose-binding protein and β -conglycinins regulate soybean seed protein content and control multiple seed traits. *Plant Physiol* 196, 1298-1321.
- Lin, Y.H., Silven, J.J.M., Wybouw, N., Fandino, R.A., Dekker, H.L., Vogel, H., Wu, Y.L., de Koster, C., Grosse-Wilde, E., Haring, M.A., Schuurink, R.C., Allmann, S., 2023. A salivary GMC oxidoreductase of *Manduca sexta* re-arranges the green leaf volatile profile of its host plant. *Nat Commun* 14, 3666.
- Liu, X., Ohta, T., Kawabata, T., Kawai, F., 2013. Catalytic Mechanism of Short Ethoxy Chain Nonylphenol Dehydrogenase Belonging to a Polyethylene Glycol Dehydrogenase Group in the GMC Oxidoreductase Family. *International Journal of Molecular Sciences* 14, 1218-1231.
- Matsui, K., Engelberth, J., 2022. Green Leaf Volatiles—The Forefront of Plant Responses Against Biotic Attack. *Plant and Cell Physiology* 63, 1378-1390.
- Phillips, D.R., Matthew, J.A., Reynolds, J., Fenwick, G.R., 1979. Partial purification and properties of a cis-3: trans-2-enal isomerase from cucumber fruit. *Phytochemistry* 18, 401-404.
- Shutov, A.D., Bäumlein, H., Blattner, F.R., Müntz, K., 2003. Storage and mobilization as antagonistic functional constraints on seed storage globulin evolution. *Journal of Experimental Botany* 54, 1645-1654.
- Smirnoff, N., Arnaud, D., 2019. Hydrogen peroxide metabolism and functions in plants. *New Phytologist* 221, 1197-1214.
- Tawfik, O.K., S., D., 2010. Enzyme Promiscuity: A Mechanistic and Evolutionary Perspective. *Annual Review of Biochemistry* 79, 471-505.
- Yoshida, H., Sakai, G., Mori, K., Kojima, K., Kamitori, S., Sode, K., 2015. Structural analysis of fungus-derived FAD glucose dehydrogenase. *Sci Rep-Uk* 5, 13498.

Reviewers' comments:

Reviewer #1 (Remarks to the Author):

The manuscript has been very nicely revised in accordance with the reviewers' comments and recommendations. I have no further concerns or suggestions for improvement. The only suggestion:

- Consistently use "lepidopteran" without capital "L" throughout the text. (Lepidoptera should be with capital).
- I would write "in planta" in italics (as it is Latin) throughout the text.

Ans: We appreciate the reviewer's time and positive assessment of our revised manuscript. We have corrected the usage of lepidopteran and Lepidoptera throughout the manuscript and highlighted these changes in red.

Regarding the suggestion to italicize in planta, we follow the Nature Portfolio style guide, which states that Latin expressions (such as in vitro and in vivo) should be formatted in roman (non-italic) type. Hence, we have kept in planta in roman font.

Reviewer #2 (Remarks to the Author):

I am 100 percent satisfied to the reviewers responses to not only my original review but also to the other reviewers comments. The authors should be commended on the thorough job they did here.

Ans: We sincerely thank the reviewer for the positive feedback and for the recognition of our efforts in addressing all reviewers' suggestions.

Reviewer #3 (Remarks to the Author):

The authors did a really great job responding to my comments (and catching an error I made regarding whether a plot had protein vs. RNA) and to those of the other reviewers. I only ask that they change Fig. 4f through g' so that there isn't red and green together otherwise they will lose colorblind readers. This is a terrific contribution to the literature.

Ans: We thank the reviewer for the positive and constructive feedback. We have changed the colors in Fig. 4f and Fig. 4g from a red-green scheme to red-blue to improve accessibility for color-blind readers. For panels 4f' and 4g', the multicolor representation is necessary to distinguish the different molecular features within the protein structure. Because these panels are also supported by textual labels identifying each feature, we have retained the current color scheme.